# C4MIP – The Coupled Climate Carbon Cycle Model Intercomparison Project: experimental protocol for CMIP6

Chris D. Jones[1], Vivek Arora[2], Pierre Friedlingstein[3], Laurent Bopp[4], Victor Brovkin[5], John Dunne[6], Heather Graven[7], Forrest Hoffman[8], Tatiana Ilyina[5], Jasmin G. John[6], Martin Jung[9], Michio Kawamiya[10], Charlie Koven[11], Julia Pongratz[5], Thomas Raddatz[5], James T. Randerson[12], Sönke Zaehle[9]

[1]Met Office Hadley Centre, Exeter, EX1 3PB, UK

[2]Canadian Centre for Climate Modelling and Analysis, Climate Research Division, Environment and Climate Change Canada

[3]College of Engineering, Mathematics and Physical Sciences, University of Exeter, Exeter, EX4 4QE, United Kingdom

[4]Laboratoire des Sciences du Climat et de l'Environnement, LSCE/IPSL, CEA-CNRS-UVSQ, Université Paris-Saclay, F-91191 Gif-sur-Yvette, France

[5]Max Planck Institute for Meteorology

[6]NOAA/GFDL, Princeton, NJ, USA

[7]Department of Physics and Grantham Institute, Imperial College London, UK

[8]Oak Ridge National Lab., TN, USA

[9]Biogeochemical Integration Department, Max Planck Institute for Biogeochemistry, D-07745 Jena, Germany

[10]Japan Agency for Marine-Earth Science and Technology, Japan

[11]Earth Sciences Division, Lawrence Berkeley National Laboratory, Berkeley, California, USA

[12]Department of Earth System Science, University of California, Irvine, USA

*Correspondence to*: C.D. Jones (chris.d.jones@metoffice.gov.uk)

**Abstract.** Coordinated experimental design and implementation has become a cornerstone of global climate modelling. Model Intercomparison Projects (MIPs) enable systematic and robust analysis of results across many models, by reducing the influence of ad-hoc differences in model set-up or experimental boundary conditions. As it enters its 6[th] phase, the Coupled Model Intercomparison Project (CMIP6) has grown significantly in scope with the design and documentation of individual simulations delegated to individual climate science communities.

The Coupled Climate-Carbon Cycle Model Intercomparison Project (C4MIP) takes responsibility for design, documentation and analysis of carbon cycle feedbacks and interactions in climate simulations. These feedbacks are potentially large and

play a leading order contribution in determining the atmospheric composition in response to human emissions of $CO_2$ and in the setting of emissions targets to stabilise climate or avoid dangerous climate change. For over a decade C4MIP has coordinated coupled climate-carbon cycle simulations, and in this paper we describe the C4MIP simulations that will be formally part of CMIP6. While the climate-carbon cycle community has created this experimental design, the simulations also fit within the wider CMIP activity, conform to some common standards including documentation and diagnostic requests and are designed to complement the CMIP core experiments known as the DECK.

C4MIP has 3 key strands of scientific motivation and the requested simulations are designed to satisfy their needs: (1) pre-industrial and historical simulations (formally part of the common set of CMIP6 experiments) to enable model evaluation; (2) idealised coupled and partially-coupled simulations with 1% per year increases in $CO_2$ to enable diagnosis of feedback strength and its components; (3) future scenario simulations to project how the Earth System will respond to anthropogenic activity over the 21st century and beyond.

This paper documents in detail these simulations, explains their rationale and planned analysis, and describes how to set-up and run the simulations. Particular attention is paid to boundary conditions, input data and requested output diagnostics. It is important that modelling groups participating in C4MIP adhere as closely as possible to this experimental design.

Keywords: Climate and Earth system modelling, CMIP6, Global Carbon Cycle, Climate Change

## 1 Introduction

Over the industrial era since about 1750, it is estimated that cumulative anthropogenic carbon emissions from fossil fuels and cement (405±20 PgC) and land use change (190±65 PgC) have been partitioned between the atmosphere (255±5 PgC), the ocean (170±20 PgC), and the terrestrial biosphere (165±70 PgC) (values to the nearest 5 PgC, from Le Quéré et al., 2015). The carbon uptake by land and ocean, since the start of the industrial era, has thus slowed the rate of increase of atmospheric $CO_2$ concentration in response to anthropogenic carbon emissions. Had the land and ocean not provided this 'ecosystem service' the atmospheric $CO_2$ concentration at present would be much higher. The manner in which the land and ocean will continue to absorb anthropogenic carbon emissions has both scientific and policy relevance. Understanding the future partitioning of anthropogenic $CO_2$ emissions into the atmosphere, land and ocean components, and the resulting climate change, accounting for biogeochemical feedbacks requires a full earth system approach to modelling the climate and carbon cycle.

The primary focus of the Coupled Climate-Carbon Cycle Model Intercomparison Project (C4MIP: http://www.c4mip.net) is to understand and quantify future century-scale changes in land and ocean carbon storage and fluxes and their impact on climate projections. In order to achieve this, a set of Earth System Model (ESM) simulations has been devised. As a consequence of the very high computational demand on modelling centres to perform a multitude of simulations for many different intercomparison studies as part of CMIP6, we have carefully chosen a minimum set of targeted simulations to achieve C4MIP goals. They comprise:

- ▪ idealized experiments which will be used to separate and quantify the sensitivity of land and ocean carbon cycle to changes in climate and atmospheric $CO_2$ concentration
- ▪ historical experiments which will be used to evaluate model performance and investigate the potential for using contemporary observations as a constraint on future projections
- ▪ future scenario experiments which will be used to quantify future changes in carbon storage and hence quantify the atmospheric $CO_2$ concentration and related climate change for a given set of $CO_2$ emissions, or, conversely, to diagnose the emissions compatible with a prescribed atmospheric $CO_2$ concentration pathway.

The simulations are designed to complement those requested in the CMIP6 DECK and the CMIP6 Historical simulation (Eyring et al., 2016). They also align closely with simulations performed as part of ScenarioMIP (O'Neill et al., 2016) by quantifying the role of carbon cycle feedbacks in the evolution of atmospheric $CO_2$ due to anthropogenic carbon emissions. Synergies with other MIPs are discussed in section 2. C4MIP simulations and analyses will play a major role contributing to the WCRP Carbon Feedbacks in the Climate System Grand Challenge (http://www.wcrp-climate.org/gc-carbon-feedbacks).

This is the third generation of C4MIP following the first coordinated experiments described in Friedlingstein et al. (2006) and the carbon cycle simulations which formed part of CMIP5 (Taylor et al., 2012).

In this paper we first briefly describe the scientific rationale and motivation for the C4MIP simulations and then carefully document the experimental protocol in section 3. Modelling groups intending to participate in C4MIP should follow the design described here as closely as possible. Particular attention should be given to the set-up of boundary conditions in terms of atmospheric $CO_2$ concentration or emissions and which aspects of the model experience changes in the fully coupled or partially coupled simulations. Output requirements (diagnostics) are also carefully documented in section 4.

Along with our science motivation (section 2) we highlight initial plans for the analyses of the carbon cycle and its interactions with the physical climate system. Modelling groups will be invited to contribute to the primary C4MIP analysis papers. We anticipate, and hope, that many further studies and analyses will also be conducted throughout the climate/carbon cycle research community and that these simulations provide a valuable resource to further carbon cycle research.

## 2 Background and science motivation

### 2.1 C4MIP history

The potential for a climate feedback on the carbon cycle whereby carbon released due to warming would further elevate atmospheric $CO_2$ and amplify climate change has been first discussed in the late 1980s-early 1990s (e.g. Lashof et al., 1989, Jenkinson et al., 1991; Schimel et al., 1994; Kirschbaum, 1995, Sarmiento and Le Quéré, 1996). On the land side, dynamic global vegetation models were used to study the impact of rising $CO_2$ and climate change on the carbon cycle (Cramer et al., 2001). There was a strong model consensus that rising $CO_2$ would stimulate additional vegetation growth and storage of carbon in terrestrial ecosystems, likewise warming climate would accelerate decomposition of dead organic matter and may also reduce vegetation productivity in some (mainly tropical) ecosystems (Prentice et al., 2001). Similarly for the ocean, there was also a model consensus that warming would lead to reduced carbon uptake (Prentice et al., 2001). This was due to both reduced solubility in warmer waters and reduced rate of transport of anthropogenic carbon to the deep ocean as a consequence of increasing stratification and shutdown of meridional overturning circulation. The processes behind the former (carbonate chemistry and solubility) were reasonably well understood (Bacastow, 1993), but the latter was much more uncertain being sensitive to the underlying ocean model circulation (Maier-Reimer et al., 1996; Sarmiento et al., 1998; Joos et al., 1999). The role of ocean biology and the buffering capacity of the ocean were also seen to be important and not well constrained or represented in models (Sarmiento and Le Quéré, 1996).

These "offline" land and ocean experiments found potentially high sensitivity of the carbon cycle to environmental forcing but were not able to simulate the full effect of this feedback onto climate. By the end of the 1990s some modelling groups

were beginning to implement interactive carbon cycle modules in their physical climate models. These early studies (e.g. Cox et al., 2000; Friedlingstein et al., 2001, Dufresne et al., 2002; Thompson et al., 2004) were able to recreate an experimental setting more like the real world where a climate change forced by anthropogenic $CO_2$ emissions would affect natural carbon sinks and stores which in turn would affect changes in atmospheric $CO_2$ and hence climate.

It soon became apparent from the first publications that there were substantial differences in the sensitivities of these new models. The desire to understand and reduce this uncertainty led to the development of a linearised feedback framework to diagnose the sensitivity of different parts of the system and their contribution to the overall feedback (Friedlingstein et al., 2003), and also of a multi-model intercomparison activity (C4MIP: Coupled Climate-carbon cycle model intercomparison, Fung et al., 2000). The result was the first C4MIP intercomparison paper, Friedlingstein et al. (2006), which quantified the feedback components across 11 models for a common $CO_2$ emissions scenario. All models agreed qualitatively that the sign of the carbon-climate feedback was positive – i.e. the interaction of the carbon cycle with climate led to reduced carbon uptake and hence an increase in atmospheric $CO_2$, which amplified the initial climate change. However, there was large quantitative model spread in the total feedback and its sensitivity components. Initial analysis of the causes of this uncertainty concluded that the land played a greater role than the ocean, in particular its sensitivity to climate. Regionally, the tropics were seen to be particularly different between models (Raddatz et al., 2007), bearing in mind that none of these models included representation of permafrost carbon. The CMIP5 experimental design for carbon cycle feedback diagnosis (Taylor et al., 2012) closely followed the C4MIP protocol. Modelling centres around the world contributed results to CMIP5 and their analysis led to many key papers including a special collection of 15 papers published in the Journal of Climate (http://journals.ametsoc.org/topic/c4mip).

The C4MIP activity under CMIP5 was central to Working Group 1 of the IPCC 5th Assessment. Several of the main findings from C4MIP studies were included in the Summary for Policymakers of WG1, such as the positive feedback between climate and carbon cycle - "Climate change will affect carbon cycle processes in a way that will exacerbate the increase of $CO_2$ in the atmosphere"; the impact of elevated $CO_2$ on ocean acidification - "Further uptake of carbon by the ocean will increase ocean acidification"; the emissions compatible with given $CO_2$ concentrations "- By the end of the 21st century, [for RCP2.6] about half of the models infer emissions slightly above zero, while the other half infer a net removal of $CO_2$ from the atmosphere"; and the very policy relevant relationship between cumulative $CO_2$ emissions and global warming - "Cumulative emissions of $CO_2$ largely determine global mean surface warming by the late 21st century and beyond".

**2.2 Key science motivation and analysis plans for C4MIP**

The key science motivations behind C4MIP are 1) to quantify and understand the carbon-concentration and carbon-climate feedback parameters which, respectively, capture the modelled response of land and ocean carbon cycle components to changes in atmospheric $CO_2$ and the associated climate change; 2) evaluate models by comparing historical simulations with

observation-based estimates of climatological states of carbon cycle variables, their variability and long-term trends; 3) to assess the future projections of the components of the global carbon budget for different scenarios, including atmospheric $CO_2$ concentration, atmosphere-land and atmosphere-ocean fluxes of $CO_2$, diagnosed $CO_2$ emissions compatible with future scenarios of $CO_2$ pathway and crucially to provide new estimates of the cumulative $CO_2$ emissions compatible with specific

climate targets. In light of the COP21 Paris agreement (https://unfccc.int/resource/docs/2015/cop21/eng/l09r01.pdf), these experiments will quantify carbon cycle feedbacks in low emissions scenarios and inform cumulative budgets consistent with a 1.5°C or 2°C stabilisation objective.

Relative to CMIP5 there are three key areas where we expect CMIP6 models to have made substantial progress and hence

may cause significant differences in the simulated response of the carbon cycle to anthropogenic forcing.

i. In CMIP5, only two participating ESMs included a land surface component (CLM4) that explicitly considered constraints of terrestrial N availability on primary production and net land carbon storage (Lindsay et al., 2014, Tjiputra et al., 2013). An increasing number of land models now include a prognostic representation of the terrestrial N cycle and its coupling to the

land C cycle (Zaehle & Dalmonech 2011). Some of these prognostic N cycle representations are expected to be used in land components of ESMs participating in CMIP6. Coupling of carbon and nitrogen dynamics changes the response of the terrestrial biosphere to global change in four ways: 1) it generally reduces the response of net primary production and carbon storage to elevated levels of atmospheric $CO_2$ because of an increasing limit of nitrogen availability for carboxylation enzymes and new tissue construction; 2) it allows for changes in plant allocation in response to changing nutrient

availability, 3) it generally decreases net ecosystem C losses associated with soil warming, because increased decomposition leads to increased plant N availability which can potentially increase plant productivity and C storage in N limited ecosystems; and 4) it alters primary production due to anthropogenic N deposition and fertiliser application, which may regionally enhance net C uptake. The magnitude of each of these processes is uncertain given strong natural gradients in the natural N availability in ecosystems and sparse ecosystem data to constrain these models (Thornton et al., 2009; Zaehle et

al., 2014; Meyerholt & Zaehle, 2015) but offline analysis of CMIP5 simulations suggests significant overestimation of terrestrial carbon uptake in models which neglect the role of nitrogen (Wieder et al., 2015; Zaehle et al., 2015). The new generation of models will provide a more comprehensive assessment of the attenuating effect of nitrogen on carbon cycle dynamics compared to CMIP5 and in particular provide a better constrained estimate of the carbon storage capacity of land ecosystems.

ii. In CMIP5, all land models used a single-layer, vertically-integrated representation of soil biogeochemistry (Luo et al., 2015). Such an approach necessarily ignores vertical variation in soil carbon turnover times, which can be very important in governing ecosystem carbon storage. This omission is most notable in the extreme case of permafrost soils, where there exists a depth at which soils remain frozen year-round and, because of the abrupt change in decomposition rates in frozen

versus unfrozen soils, otherwise highly decomposable carbon can be preserved indefinitely until it is thawed. The majority of global soil carbon is in permafrost-affected ecosystems, which creates the possibility for permafrost climate feedbacks (Burke et al., 2013). Some of the models in CMIP6 are expected to include representation of permafrost soil carbon dynamics, either explicitly by representing soil biogeochemistry along the full soil depth axis (Koven et al., 2014), or by means of reduced-complexity methods to incorporate permafrost dynamics. AR5 concluded that permafrost carbon release was likely, and therefore would increase the climate-carbon cycle feedback, but with low confidence in the magnitude (Ciais et al., 2013). Assessing the role of this process in governing fully-coupled climate feedbacks will be an important contribution to CMIP6.

iii. Representation of ocean dynamics in the ESMs is another important constraint affecting the oceanic carbon uptake and storage. There is evidence that by shifting to an eddy-permitting grid configuration of the ocean general circulation model, the representation of some key features of oceanic circulation such as the interior water mass properties and surface ocean current systems are improved (Jungclaus et al., 2013). The increased horizontal resolution of the underlying ocean model has a positive impact on the performance of the marine biogeochemistry model in the deeper layers (Ilyina et al., 2013). Spatial resolution of some ESMs is expected to increase as they move into CMIP6. The increased resolution of the oceanic components of the ESMs is expected to have some explicit advantages for projections of the oceanic carbon uptake. First, it allows us to estimate the role of previously unresolved small-scale ocean hydrodynamical process on projections of marine biogeochemistry. Second, by improving the representation of coastal processes and ocean-shelf exchange, their contribution to the global carbon cycle can be assessed.

## 2.2.1. Carbon cycle feedback parameters

The first key motivation for C4MIP is to document the changes in magnitude of the feedback parameters that characterize the response of the carbon cycle and their spread across models through time. In this respect, C4MIP aims to calculate the magnitude of the carbon-concentration and carbon-climate feedbacks in a manner similar to Friedlingstein et al. (2006) or Arora et al. (2013) and as discussed in Section 3.1 using results from the idealized 1% per year increasing $CO_2$ experiments.

The 1pctCO2 experiment has gained recognition as a standard CMIP simulation and it is one of the DECK simulations for CMIP6 (Eyring et al., 2016). The 1pctCO2 experiment is now routinely used to characterize the transient climate response (TCR) defined as the change in globally-averaged near-surface air temperature at the time of $CO_2$ doubling as well as the transient climate response to cumulative emissions (TCRE) defined as change in globally-averaged near-surface air temperature per unit cumulative $CO_2$ emissions at the time of $CO_2$ doubling (Gillett et al., 2013). In addition, since the 1pctCO2 simulation does not include the confounding effects of changes in land use, non-$CO_2$ greenhouse gases, and aerosols it provides a clean controlled experiment with which to compare carbon–climate interactions across models. Its backwards compatibility enables direct comparison of models with previous generations, which has been hindered

previously as the scenario-dependence of the feedback metrics has prevented a like-for-like comparison (Gregory et al., 2009).

C4MIP will use partially coupled simulations to isolate and quantify the sensitivity of carbon cycle components to climate and $CO_2$ separately and also the potentially large non-linear combination of these two components (Gregory et al., 2009; Schwinger et al., 2014). Simulations with only the carbon cycle model components experiencing rising $CO_2$ (BGC-coupled) and the radiation model components experiencing rising $CO_2$ (RAD-coupled) are used to quantify the carbon-concentration and carbon-climate feedbacks. Spatial patterns of these metrics can also be calculated (e.g. Roy et al., 2011; or Fig. 6.22 of the last IPCC WG1 assessment report Ciais et al., 2013) to establish areas of model agreement or disagreement.

### 2.2.2. Evaluation of global carbon cycle models

The historical simulations will be used for evaluation of the components of the carbon cycle (ocean and terrestrial carbon fluxes, anthropogenic carbon storage in the ocean, atmospheric $CO_2$ growth rate and variability). ESMs have increased rapidly in complexity but evaluation has not kept pace. Some evaluation of the carbon cycle was already performed in CMIP5 (e.g. Anav et al., 2013, Bopp et al., 2013; Hoffman et al., 2014), highlighting significant biases in key quantities in many ESMs. There is increasing need to develop evaluation techniques and activities, applied consistently and routinely across models, at both fine scales (process-level, "bottom-up" evaluation) and large scales (system-level, "top-down" evaluation"), as well as using complementary data streams relating to (bio)physical and biogeochemical processes to evaluate the ensemble of simulated processes (e.g. Luo et al., 2012; Foley et al., 2013).

Evaluation of ocean carbon cycle components of ESMs has been classically based on the use of the monthly surface $pCO_2$ climatology of Takahashi et al. (2009), derived from more than 3 million in-situ ocean $pCO_2$ measurements, as in Pilcher et al. (2015) for an evaluation of $pCO_2$ seasonality of the CMIP5 ESMs. This evaluation is complemented by the use of additional climatological gridded products, as in Anav et al. (2013), with model-data comparison for related physical variables (e.g. mixed layer depth) or biological variables (e.g. net primary production). In the past few years, ESM evaluation has extended in many directions, making use of advanced observation-based gridded products (e.g. the three-dimensional distribution of anthropogenic carbon in the ocean from Khatiwala et al. (2013) ) and ocean databases with millions of in-situ point measurements (e.g. with the Surface Ocean $CO_2$ Atlas (SOCAT) as in Tjiputra et al. (2014) for CMIP5 ESMs), or developing new techniques for model-data comparisons (e.g. water-mass framework; Iudicone et al., 2011).

In the coming years, the increasing complexity of marine biogeochemical schemes used in ESMs will call for more advanced model-data comparison strategies. These will include the use of new data sets, such as biomass data for plankton functional types (MAREDAT, Buitenhuis et al., 2013) or ocean distribution of the micro-nutrient iron (Tagliabue et al., 2012).

Evaluations of land surface components of ESMs have often used gridded flux products (e.g. Bonan et al., 2011; Anav et al., 2013; Piao et al., 2013) obtained by extrapolating the FLUXNET measurement network of biosphere-atmosphere exchanges (e.g. Jung et al., 2011), for instance to constrain modelled spatial and seasonal distribution of gross primary production (GPP). Such products are convenient for such model evaluations because those are available at a resolution comparable to that of the models and because they retain the pertinent patterns of the observed fluxes while abstracting from measurement noise, local site representativeness and other possible site-specific features. Yet it is important to bear in mind the limitations of the 'upscaled' flux and stock products and to tailor the model evaluation to robust patterns that the individual products are ideally suited for. Insights may also be gained from evaluation of functional patterns and sensitivities to certain climate forcing variables. For example the spatial sensitivity of GPP with mean annual precipitation in the water-limited domain, and the temperature sensitivity of ecosystem respiration (Mahecha et al., 2010).

While data-model comparisons of fluxes are important, they alone cannot constrain longer-term dynamics and associated climate-carbon cycle feedbacks. In addition, consideration of residence times is crucial, which together with carbon fluxes jointly determine the stores. Analysis of CMIP5 ESMs revealed unacceptably large errors in land carbon stores (both in living biomass and soil organic matter) (Anav et al., 2013). Future simulation results were found to depend on the initial conditions as well as the model sensitivity to changes (Todd-Brown et al., 2014) and therefore better evaluation and constraint of carbon stores is seen as vital. Xia et al. (2013) showed the importance of residence time in determining carbon stores and Carvalhais et al. (2014) showed the mismatch between CMIP5 ESMs and an observationally-derived dataset of land-carbon residence times. As more observations become available (Saatchi et al., 2011; Baccini et al., 2012; Avitabile et al., 2015; FAO 2012; Batjes et al., 2012; Hengl et al., 2014) as well as data constrained products such as residence time (Bloom et al., 2016), we stress the importance of rapid development and application of evaluation techniques to ESMs.

Carbon isotopes (carbon-13 and carbon-14) provide unique insights into the mechanisms and timescales of carbon cycling. Differences between the isotopic fractionation of carbon from dissolution in the ocean and from photosynthetic assimilation on land have enabled atmospheric observations of the $^{13}C/^{12}C$ ratio ($\delta^{13}C$) in atmospheric $CO_2$ to be used in differentiating land and ocean carbon fluxes (Ciais et al., 1995; Joos et al., 1998; Rubino et al., 2013). The perturbation of the $^{14}C/C$ ratio ($\Delta^{14}C$) in atmospheric $CO_2$ from nuclear weapons testing in the 1950s and 60s has provided a valuable tracer of carbon turnover rates in terrestrial carbon pools (Trumbore, 2000; Naegler and Levin, 2009), and the rates of air-sea exchange and ocean mixing, including constraints on ocean $CO_2$ uptake (Matsumoto et al., 2004; Sweeney et al., 2007; Graven et al. 2012). Integration of carbon isotopes into ESMs is an emerging activity and we request the reporting of carbon isotopic variables for the first time in C4MIP. Carbon isotopes are also included in OMIP (Orr et al., 2016). ESMs that simulate carbon isotopes are requested to report fluxes and stocks of carbon isotopes in their land and ocean components. This will enable comparison between models currently simulating carbon isotopes and their evaluation by observations, as well as encourage

future development of carbon isotopes in ESMs. Simulation of carbon isotopes in C4MIP is expected to provide novel insights on ocean mixing and air-sea exchange, marine ecosystem change, plant water use efficiency and stomatal closure especially during drought periods, and terrestrial carbon residence times.

Historical simulations will be needed to explore potential emergent constraints from observations on the future response of the carbon cycle, with a particular focus on carbon cycle feedbacks. Recent studies showed the potential of observed interannual $CO_2$ variability to constrain the future tropical land carbon cycle sensitivity to climate change (Cox et al., 2013; Wenzel et al., 2014).

In the same way that earth system modelling has become an internationally collaborative activity involving shared expertise and development of tools, we also expect that evaluation techniques will evolve in this way. Community evaluation activities such as ILAMB (http://www.ilamb.org/) and ESMValTool (Eyring et al., 2016b) look likely to become increasingly useful for addressing the complexities of multi-model ESM evaluation.

### 2.2.3. Future projections of the components of the global carbon budget

While idealized experiments are useful for intercomparison of climate-carbon interactions across multiple models, they do not take into account the effect of non-$CO_2$ GHGs, aerosols and land use change, all of which affect the behaviour of the carbon cycle in the real world. In contrast, the scenarios considered by the ScenarioMIP are internally coherent in all aspects of anthropogenic forcings. Within each socio-economic storyline, changes in fossil fuel $CO_2$ emissions are consistent with those in aerosols emissions, N deposition, and changes in land use areas, all of which are based on plausible assumptions of 20 demographic and economic development in the future. This plausibility is of special interest to policymakers. Scenarios also indicate the range of possible future developments and opportunities for mitigation and adaptation; this information is used widely in climate impact analyses.

The scenario simulations, therefore, provide more realistic conditions compared to the idealised 1% experiments due to their 25 plausibility of anthropogenic forcings as well as the longer time scale over which the $CO_2$ increase occurs. Since shared socio-economic pathway (SSP) scenarios include all forcings, their climate and biogeochemical effects are able to influence the atmosphere-surface carbon exchange for both land and ocean components. Emission-driven historical and the future SSP5-8.5 simulations replicate a more realistic model setting where ESMs are directly forced by anthropogenic $CO_2$ emissions, allowing for the carbon cycle feedbacks to impact on atmospheric $CO_2$ and simulated climate change. These will 30 be compared with the concentration-driven equivalents in ScenarioMIP and additionally will form a baseline control experiment for analysis of alternative future land use scenarios in LUMIP (Lawrence et al., 2016).

The proposed biogeochemically-coupled versions of the historical and future SSP5-8.5 in Section 3.1, in which $CO_2$ induced warming is not accounted for, when compared to their fully-coupled versions will allow us to investigate the effect of $CO_2$ induced warming on atmosphere-land and atmosphere-ocean $CO_2$ fluxes over the 20[th] and 21[st] century and beyond (Randerson et al., 2015). An important objective with these simulations will be to identify how land and ocean contributions to feedbacks and compatible emissions evolve century by century from sustained increases in ocean heat content and thawing of permafrost soils.

ScenarioMIP (O'Neill et al., 2016) acknowledges scientific and policy interest in a scenario with a substantial overshoot in radiative forcing during the 21[st] century. As such they include a tier-2 concentration-driven scenario called SSP5-3.4-OS, an overshoot pathway, which follows SSP5-8.5 up to 2040 followed by aggressive mitigation to reduce emissions to zero by about 2070 followed by substantial negative global emissions thereafter. The carbon cycle response to peak-and-decline $CO_2$ levels is likely to differ from the response to continued strong increases in $CO_2$. The 21[st] century airborne fraction from CMIP5 models varied substantially between RCPs, with RCP2.6 in particular having a much lower airborne fraction than the 20[th] century or other RCPs (Jones et al., 2013). However, to date there have been no coordinated experiments to quantify the carbon-cycle feedback components in such a scenario. Hence for C4MIP we include a biogeochemically coupled ("BGC") simulation of the SSP5-3.4-OS scenario.

## 2.3 Links to and requirements from other MIPs

The Ocean Model Intercomparison Project (OMIP, Griffies et al., 2016; Orr et al., 2016) will provide a baseline for assessment of ocean component model biogeochemical and historical carbon uptake fidelity. Ocean carbon cycle analysis has previously been conducted under the OCMIP intercomparison (Orr et al., 2001). In response to the WGCM request, the OMIP and OCMIP have been merged under the OMIP umbrella. One main objective of OMIP is to coordinate CMIP6 ocean diagnostics including ocean physics, inert chemical tracers, and biogeochemistry for all CMIP6 simulations that include an ocean component. The second objective is to perform a global ocean/sea-ice simulation forced with common atmospheric data sets. In this way, ocean models including online biogeochemistry components will be part of "Path-II" simulation, (whereas "Path-I" is designated to models without the biogeochemistry). Within OMIP, ocean-only simulations will be performed as described in Griffies et al. (2016).

Analysis of changes in terrestrial carbon stocks for historical and future scenarios as result of changes in atmospheric $CO_2$, climate, and land-use and land-use-induced land cover change (LULCC) will be done in coordination with LUMIP (Lawrence et al., 2016). The emission-driven future scenario performed within C4MIP serves as control simulation for LUMIP. By replacing the LULCC forcing of SSP5-8.5 by the one from SSP1-2.6 under otherwise identical forcings the effect of LULCC can thus be isolated. This also implies that output provided for the emission-driven simulation should

account for the additional requirements of LUMIP such as tile-level reporting of variables. Offline land-surface process studies form part of LS3MIP (van den Hurk et al., 2016) and offline simulations to quantify the contemporary land carbon budget are performed under the TRENDY intercomparison (Sitch et al., 2015).

The scientific scope of the Detection and Attribution intercomparison (DAMIP) includes attempting some observational constraint on the transient climate response to cumulative carbon emissions (TCRE: Gillett et al., 2016), whose assessment is also an important target of C4MIP. Collaborative opportunities exist between C4MIP and DAMIP for analyses of TCRE with C4MIP covering carbon cycle aspects of the historical runs. Furthermore, results from DAMIP analysis runs will provide insights on the mechanism of fluctuations of past $CO_2$ growth rate. Synergies also exist between DAMIP and
LUMIP, and also RFMIP, regarding the biophysical effects of land-use change.

### 3. C4MIP Experiments

### 3.1 Overview of simulations and their purpose

The C4MIP protocol for CMIP6 builds on DECK and historical CMIP6 simulations which are documented in detail in (Eyring et al., 2016). The following experiments are not formally C4MIP simulations but are considered pre-requisite
simulations for C4MIP analyses:

- CMIP DECK pre-industrial control simulation (piControl), with specified $CO_2$ concentration ("concentration driven")
- CMIP DECK pre-industrial control simulation (esm-piControl), with interactively simulated atmospheric $CO_2$ ("emissions driven", but with zero emissions)
- CMIP DECK 1% per year increasing $CO_2$ simulation (1pctCO$_2$) initialized from pre-industrial $CO_2$ concentration until quadrupling. In C4MIP terminology this is "fully-coupled" meaning that both the model's radiation and carbon cycle components see the increasing $CO_2$ concentration.
- CMIP6 concentration-driven historical simulation for 1850-2014 (historical).
- CMIP6 emissions-driven historical simulation with interactively simulated atmospheric $CO_2$ (esm-hist) forced by
anthropogenic emissions of $CO_2$. Other forcings such as non-$CO_2$ GHGs, aerosols, and land-cover change are being prescribed as in the CMIP6 concentration-driven historical simulation.

These simulations are documented in detail in Eyring et al. (2016), but here we emphasise some carbon-cycle specific aspects and requirements.

The simulations specifically identified as C4MIP simulations are separated into two tiers. We require only a minimalistic two experiments for C4MIP Tier-1 analysis. These are:

- Biogeochemically-coupled version of the 1% per year increasing $CO_2$ simulation (1pctCO2-bgc)
- Emissions-driven future scenario based on the SSP5-8.5 scenario (esm-ssp585).

The rationale for these two required simulations is that they form a minimum set of outputs required to quantify the climate-carbon cycle feedback in a model and to simulate the full effects of this feedback on future climate under a high-end
emissions scenario. The emissions scenario also provides a control for the LUMIP esm-ssp585-ssp126Lu simulation.

Further simulations are then requested under C4MIP Tier-2 which allow a more complete investigation of the feedback components, their non-linearities, their sensitivity to nitrogen limitations (if included in the model) and the role of their effects on future scenarios including sustained $CO_2$ increases and a peak-and-decline in forcing. It is highly desirable that as
many of these as possible are performed to accompany the tier-1 simulations. They are divided into two categories:

i. idealised simulations
- Radiatively-coupled (RAD) version of the 1% per year increasing $CO_2$ simulation (1pctCO2-rad),
- Fully-coupled (COU) 1% per year increasing $CO_2$ simulation with nitrogen deposition (1pctCO2Ndep), and
- Biogeochemically-coupled (BGC) version of the 1% per year increasing $CO_2$ simulation with nitrogen deposition (1pctCO2Ndep-bgc).

ii. scenario simulations
- Biogeochemically-coupled version of the concentration-driven historical CMIP6 simulation (hist-bgc),
- Biogeochemically-coupled version of the concentration-driven future SSP5-8.5 scenario (ssp585-bgc),
- Biogeochemically-coupled version of the concentration-driven future extension of the SSP5-8.5 scenario (ssp585-bgcExt)
- Biogeochemically-coupled version of the concentration-driven future SSP5-3.4-over scenario (ssp534-over-bgc),
- Biogeochemically-coupled version of the concentration-driven future extension of the SSP5-3.4-over scenario (ssp534-over-bgcExt),

Note that 1pcCO2Ndep and 1pcCO2Ndep-bgc are only applicable to models whose simulation will be affected by the deposition of reactive nitrogen either due to terrestrial or marine nitrogen cycle effects on carbon fluxes and stores. Similarly, the biogeochemically forced scenario simulations (ssp585-bgc and ssp534-over-bgc) are only required if the coupled ScenarioMIP counterpart has been performed (ssp585 and ssp534-over). If computing resource limits the number of
simulations performed we recommend prioritising ssp585-bgc over ssp534-over-bgc.

The simulations required for C4MIP are summarised in table 1 and the $CO_2$ concentration is shown schematically in figure 1 in the context of the CMIP6 DECK, historical simulations and ssp585 future scenario which is a Tier 1 experiment of the

ScenarioMIP (O'Neill et al., 2016). Table 2 shows the main simulations from other MIPs which form crucial counterparts to C4MIP simulations. The rest of this section documents detailed instructions on how to set-up and perform the C4MIP simulations. Detailed definitions of the output requirements are listed in section 4.

| Category | Type of Scenario | Emission or concentration driven | Coupling mode | Simulation years | Short name |
|---|---|---|---|---|---|
| **Tier 1** | | | | | |
| **1%BGC** | Idealised 1% per year $CO_2$ only, BGC mode | C-driven | $CO_2$ affects BGC | 140 | 1pctCO2-bgc |
| **SSP5-8.5** | SSP5-8.5 up to 2100 | E-driven | Fully coupled | 85 | esm-ssp585 |
| **Tier 2** | | | | | |
| **1%RAD** | Idealised 1% per year $CO_2$ only, RAD mode | C-driven | $CO_2$ affects RAD | 140 | 1pctCO2-rad |
| **1%COU-Ndep** | Idealised 1% per year $CO_2$ only, fully coupled, increasing N-deposition | C-driven | Fully coupled | 140 | 1pctCO2Ndep |
| **1%BGC-Ndep** | Idealised 1% per year $CO_2$ only, BGC mode, increasing N-deposition | C-driven | $CO_2$ affects BGC | 140 | 1pctCO2Ndep-bgc |
| **Hist/SSP5-8.5-BGC** | Historical+SSP5-8.5 up to 2300, BGC mode | C-driven | $CO_2$ affects BGC | i. 165<br>ii. 85<br>iii. 200 | hist-bgc, ssp585-bgc and ssp585-bgcExt |
| **SSP5-3.4-Overshoot-BGC** | SSP5-3.4-OS up to 2300 in BGC mode | C-driven | $CO_2$ affects BGC | i. 60 (from 2040-2100)<br>ii. 200 | ssp534-over-bgc, ssp534-over-bgcExt |

*Table 1. Summary of the C4MIP tier-1 and tier-2 simulations. Simulations can be "concentration driven" or "emissions driven" as described in the text. Coupling mode refers to which model components see changes in atmospheric $CO_2$.*

| Type of simulation | Simulation name | Owning MIP | notes |
|---|---|---|---|
| **Control** | | | |
| | piControl | DECK | Prescribed pre-industrial $CO_2$ concentration |
| | esm-piControl | DECK | Prognostically simulated atmospheric $CO_2$ concentration. Required if performing any emissions-driven simulations for C4MIP |
| **Idealised** | | | |
| | 1pctCO2 | DECK | Forms essential counterpart for C4MIP BGC and RAD 1% simulations |
| **Historical** | | | |
| | historical | CMIP6 Historical | |
| | esm-hist | CMIP6 Historical | Prognostically simulated atmospheric $CO_2$ concentration. Required if performing any emissions-driven simulations for C4MIP. Provides starting point for C4MIP emissions-driven SSP5-8.5 |
| **Future scenarios** | | | |
| | ssp585, ssp585ext | ScenarioMIP | Essential counterpart for SSP5-8.5-BGC de-coupled simulation and its extension to 2300 |
| | ssp534-over, ssp534-over -ext | ScenarioMIP | Essential counterpart for SSP5-3.4-over-bgc de-coupled simulation and its extension to 2300. Branches from SSP5-8.5 at 2040. |
| | esm-ssp585-ssp126Lu | LUMIP | Same as esm-ssp585 except uses SSP1-2.6 land use (afforestation scenario). |

*Table 2. Summary of key simulations from CMIP6 DECK, Historical or other MIPs on which C4MIP analysis will rely. The emissions-driven control and historical runs in particular are entry card requirements for C4MIP.*

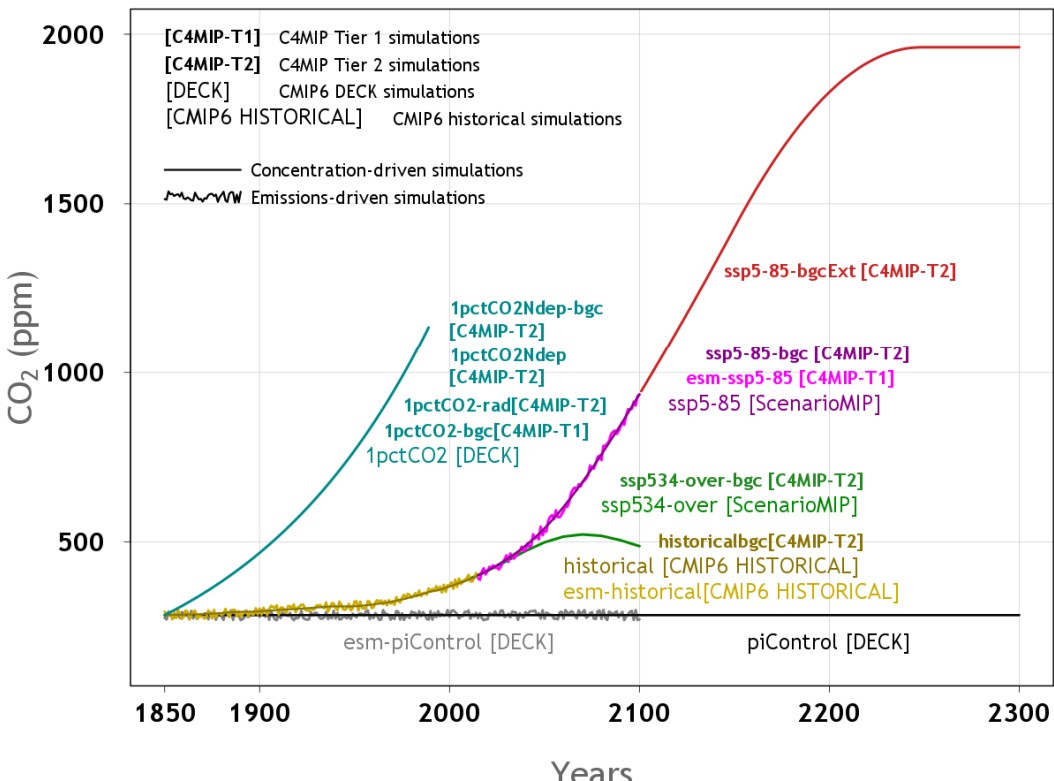

*Figure 1: Relation of C4MIP simulations to CMIP6 DECK and historical simulations and the ssp585 and ssp5-34-over future scenario simulation proposed for the ScenarioMIP. Note that at the time of preparing this manuscript the details of the SSP5-3.4-OS-Ext extension to 2300 are not available, hence it could not be included in the figure, but it is still requested as a C4MIP tier-2 simulation.*

**3.2 Experimental details**

**3.2.1 Model requirements and spin-up**

To participate in C4MIP a climate model must have the capability to run with an interactive carbon cycle. This means it must simulate both terrestrial and marine carbon cycle processes, and it must simulate the exchange of $CO_2$ between the land/ocean and the atmosphere in order to prognostically simulate the evolution of atmospheric $CO_2$. Some C4MIP simulations prescribe a concentration of $CO_2$ in the atmosphere as a boundary condition and simulate the changes in carbon fluxes and stores in response. Other simulations prescribe emissions of $CO_2$ to the atmosphere (from human activity) as an

external forcing and require the model to also simulate the evolution of atmospheric $CO_2$. A model must be able to run in both these configurations in order to perform the C4MIP simulations. The evolution of atmospheric $CO_2$ concentration can be simulated by assuming that $CO_2$ is completely well mixed with the same globally-averaged concentration everywhere in space or by transporting $CO_2$ as a three-dimensional tracer. This choice is up to the modelling groups. Throughout this document we refer to the former – prescribing atmospheric $CO_2$ concentration as a boundary condition – as a "concentration driven" simulation, and the latter – prescribing emissions and in turn simulating the $CO_2$ concentration – as an "emissions driven" simulation. IPCC AR5 WG1 Ch.6 Box 6.4 described the use of these configurations in some detail (Ciais et al., 2013). Figure 6.4 from that Box is reproduced here for reference (figure 2). Although the same terminology (concentration-driven or emissions-driven) can be applied to aerosols or non-$CO_2$ GHGs this paper focuses only on $CO_2$.

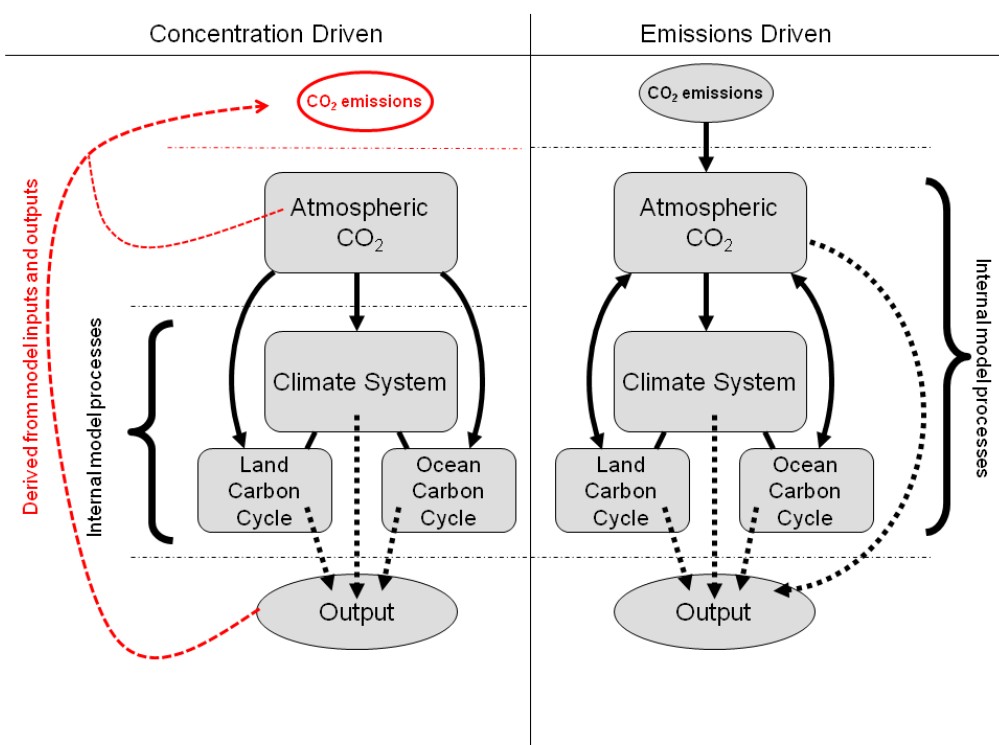

*Figure 2. Schematic representation of carbon cycle numerical experimental design. Concentration-driven (left) and emissions-driven (right) simulation experiments make use of the same Earth System Models (ESMs), but configured differently. Concentration-driven simulations prescribe atmospheric $CO_2$ as a pre-defined input to the climate and carbon cycle model components. Compatible emissions can be calculated from the output of the concentration-driven simulations. Emissions-driven simulations prescribe $CO_2$ emissions as the input, and atmospheric $CO_2$ is an internally calculated state variable within the ESM. Adapted from Ciais et al. (2013). Solid arrows depict internal data flow within the model, dashed arrows depict data output from the model.*

Before beginning the simulations described below, a model must be spun-up to eliminate any long-term drift in carbon stores or fluxes. Indeed, it has been shown recently that the large diversity in spin-up protocols used for marine biogeochemistry in CMIP5 ESMs contribute to large model-to-model differences in simulated fields, and that drifts have potential implications on model performance assessments in addition to possibly aliasing estimates of climate change impacts (Séférian et al., 2015). Separate spin-up simulations should be performed for both concentration-driven and emission-driven configurations. There are many possible techniques to ensure that a model's carbon fluxes and pools exhibit minimal drift. These include simply performing very long simulations, running components offline from the coupled system, numerical acceleration techniques or semi-analytical schemes such as described by Xia et al. (2012). The choice of technique is up to the modelling groups and there is no requirement to submit data from the spin-up period, but a proper documentation of the spin-up technique and duration is required. The test of whether a model is spun-up properly and exhibits minimal drift will be based on the performance of the piControl simulation. It is suggested that the model first be spun-up in concentration-driven mode and this state can be used as an initial basis for the emission-driven spin-up.

Our definition of an acceptably small drift in a properly spun-up model is that land, ocean and atmosphere carbon stores each vary by less than 10 PgC/century (i.e., a long-term average $\leq 0.1$ PgC/year). This is broadly equivalent to an atmospheric $CO_2$ drift of less than about 5 ppm/century. We suggest that a drift smaller than this value is highly desirable but this value is a guideline. Exceeding this drift in the control run may preclude a model from being included in a C4MIP analysis, but we would expect that decision to be made on a case-by-case basis. For example, a large ocean drift in a concentration-driven experiment may not preclude analysis of land carbon fluxes and vice-versa. We also stress that being within these drifts is a minimum but not necessarily sufficient quality condition. Regional patterns and drifts of stores and fluxes will also be assessed and depending on the analysis may preclude inclusion of a given model's results.

For simulations of carbon isotopes, spin-up times of many thousands of years or the use of an equivalent fast spin-up technique may be required to eliminate drift, particularly for carbon-14 in ocean carbon and soil carbon. The spin-up technique is left to the modellers' discretion.

### 3.2.2 DECK piControl and Historical

The pre-industrial control run (piControl) is a required simulation of the CMIP DECK, and a pre-requisite simulation for participating in C4MIP. The run begins from a spun-up state as described above and all forcings should continue to be applied as per the spin-up. The global land and ocean carbon stores should not drift by more than 10 PgC/century each. The length of the pre-industrial control run should be at least equal to any simulation for which it will serve as the control simulation thereby allowing correction for model drift. The piControl run must be run for both concentration-driven and emission-driven configurations of the model. In both cases all forcings should be held constant at pre-industrial levels as

described in the CMIP DECK documentation. The only difference between concentration-driven and emission-driven control runs is that the emission-driven simulation simulates atmospheric $CO_2$ internally in response to natural fluxes of carbon from land and ocean, while in the concentration-driven case atmospheric $CO_2$ concentration is specified. No anthropogenic fossil-fuel emissions of $CO_2$ should be applied to the model during this control run, and fixed pre-industrial

land-use should be imposed. The simulated atmospheric $CO_2$ in esm-piControl should therefore remain stable, with drifts below 5 ppm/century.

The CMIP6 Historical run, a CMIP6 required simulation, must be performed in both concentration-driven and emission-driven configurations for participation in C4MIP. It is expected that the historical simulation would begin from the same

starting point as the pre-industrial control run (figure 3). This nominally is set as 1 January 1850. We note though that this neglects the small but non-zero effect of pre-1850 land-use changes (see e.g., Pongratz et al., 2009; Sentman et al., 2011). Some modelling groups might therefore opt for an earlier starting date or perform additional offline land-surface simulations in order to account for pre-1850 land cover change. This would mean though that the control and historical simulations begin from different states and with different trends and this should therefore be very clearly documented. The protocol for the

historical simulation is documented in detail in the CMIP6 paper (Eyring et al., 2016). Here we stress the need for the emission-driven historical run (esm-hist) to also be performed as an "entry card" for C4MIP. The only difference between concentration-driven and emission-driven simulations is the treatment of atmospheric $CO_2$. All other forcings must be identical in both simulations. The concentration-driven simulation will use historical atmospheric $CO_2$ concentration provided by CMIP6.

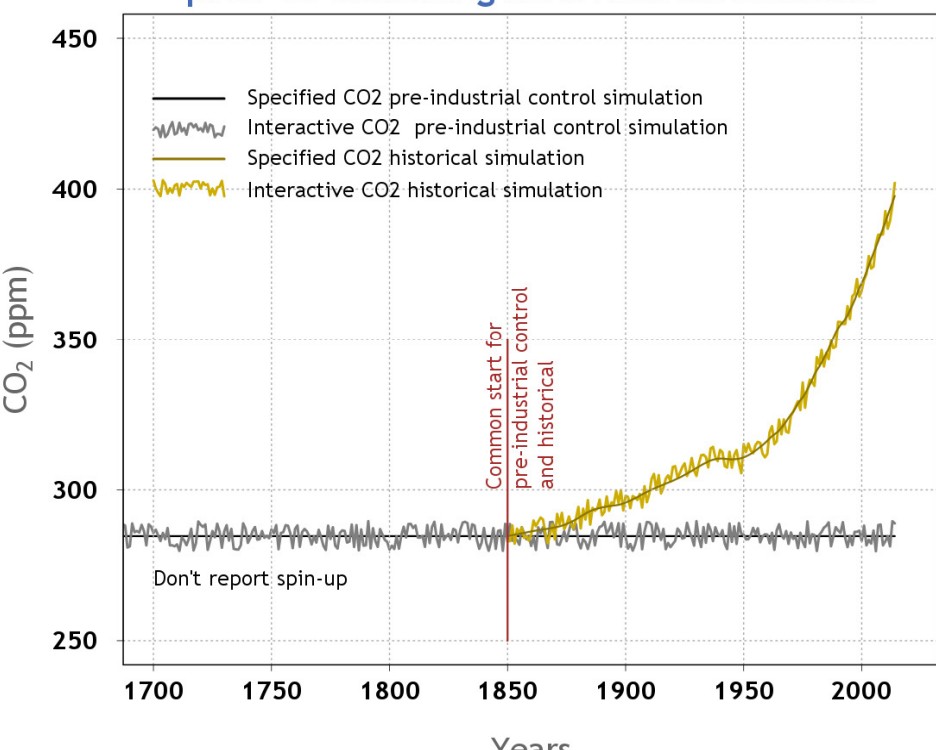

*Figure 3. Schematic representation of model spin-up followed by control and historical simulations through 2014. The interactive CO₂ pre-industrial control should ideally have a drift of less than 5 ppm/century.*

The emission-driven simulation will use anthropogenic $CO_2$ emissions documented here. Model groups have a choice over the treatment of land-use forcing as described below.

- Fossil fuel emissions. CMIP6 will provide gridded, annual $CO_2$ emissions from burning of fossil fuels, from the beginning of 1850 to the end of 2014 for the historical simulation and through to the end of 2100 for ssp5-8.5. See section 3.3.1.

- Land-use carbon emissions. There are 2 allowable options:

      o   if possible, drive the model with the CMIP6 land-use forcing (Hurtt et al., 2016, in prep; http://luh.umd.edu/~LUH2/LUH2_v1.0h/) and the model simulates its own $CO_2$ emissions (including both from deforestation and uptake from regrowth) to/from the atmosphere as an internal process. In this case the only external input of carbon to the system is fossil fuel emissions.

o   If that is not possible for the model, then C4MIP will provide land-use carbon emissions. See section 3.3.1.

### 3.2.3 Idealised 1% simulations

A concentration-driven simulation with a 1% per year increase in atmospheric $CO_2$ concentration beginning from pre-industrial is a required simulation of the DECK. In C4MIP there are further variants of this 1% simulation designed to quantify the concentration-carbon and climate-carbon feedback parameters (Friedlingstein et al., 2006; Arora et al., 2013).

The tier-1 C4MIP simulation 1pctCO2-bgc requires the simulation to be repeated but with a change to the model set-up such that only the model's carbon cycle components (both land and ocean) responds to the increase in $CO_2$, whereas the model's radiation code uses a constant, pre-industrial concentration of $CO_2$. This simulation was previously known as "Uncoupled" in Friedlingstein et al. (2006), and was re-named "Biogeochemically coupled" by Gregory et al. (2009). All other forcings

must be identical to the DECK 1pctCO2 simulation.

A tier-2 C4MIP simulation 1pctCO2-rad is the counterpart of 1pctCO2-bgc. It requires the simulation to be repeated but with a change to the model set-up such that only the model's radiation code sees the increase in $CO_2$ and the model's carbon cycle components (both land and ocean) see a constant, pre-industrial concentration of $CO_2$. This simulation was not performed in

Friedlingstein et al. (2006), and was termed "Radiatively coupled" by Gregory et al. (2009). All other forcings must be identical to the DECK 1pctCO2 simulation. Although this simulation is in tier-2 we strongly encourage all modelling groups to perform it as the non-linearities of biogeochemical and radiative response can be large (see e.g. Schwinger et al., 2014).

For models with a nitrogen cycle there are two further 1% simulation variants requested as C4MIP tier-2: 1pctCO2Ndep,

1pctCO2Ndep-bgc. These can be run if the model includes either land- or marine nitrogen cycle in a way that changes carbon uptake and storage. If the input of reactive nitrogen to the model will not affect the carbon cycle, then there is no need to perform these simulations. If changes in nitrogen deposition will affect either land or ocean carbon uptake then these simulations are requested. The 1pctCO2Ndep and 1pctCO2Ndep-bgc parallel the 1pctCO2 and 1pctCO2-bgc simulations but with the addition of a time varying deposition of reactive nitrogen (see section 3.3.3).

### 3.2.4 Scenario simulations

Concentration-driven scenario simulations which follow on from the end of the concentration-driven historical simulation are performed under ScenarioMIP. In C4MIP we request simulations which complement some of these.

Under C4MIP tier-1 we request an emission-driven esm-ssp585 simulation which parallels the ScenarioMIP concentration-

driven SSP-5-8.5 simulation. This simulation should begin from the end point of the emissions-driven historical simulation (January 1[st] 2015). As with the historical simulation the only difference from the concentration-driven counterpart should be the treatment of atmospheric $CO_2$, which is simulated within the model driven by prescribed emissions. SSP8.5 gridded

fossil-fuel emissions will be provided as will SSP8.5 land-use forcing and land-use $CO_2$ emissions. Models should implement these in the scenario run in exactly the same manner as they did in the emission-driven historical simulation.

Under C4MIP tier-2 we also request a biogeochemically-coupled ("BGC") version of the concentration-driven SSP5-8.5, ssp585-bgc and ssp585-bgcExt. As with the 1pctCO2-bgc simulation, this run should be performed with only the carbon cycle components (land and ocean) seeing the prescribed increase in atmospheric $CO_2$. The model's radiation scheme should see fixed pre-industrial $CO_2$. All other non-$CO_2$ forcings should be applied in an identical way to the ScenarioMIP SSP5-8.5 and SSP5-8.5ext simulations. If possible this simulation should be extended to 2300, as should its counterpart from ScenarioMIP, as one of the priority focus areas for analysis is on long-term processes such as ocean carbon and heat uptake and permafrost loss (e.g., Randerson et al., 2015).

## 3.3 Forcings and inputs

### 3.3.1 $CO_2$ concentrations and anthropogenic $CO_2$ emissions

For concentration driven simulations, atmospheric $CO_2$ should be prescribed as a globally well mixed value provided by CMIP6. The CMIP6 paper (Eyring et al., 2016) and a range of papers in the GMD CMIP6 special issue will document the forcings in more detail. The data will be made available from the CMIP6 and PCMDI webpages (http://www.wcrp-climate.org/wgcm-cmip/wgcm-cmip6, https://pcmdi.llnl.gov/search/input4mips). For emissions driven simulations, atmospheric $CO_2$ should be simulated prognostically by the model. External boundary conditions of anthropogenic $CO_2$ emissions will be provided and should be used as follows:

- in esmPIcontrol, the emissions-driven control run, atmospheric $CO_2$ should be simulated by the model but no external emissions should be added during this simulation

- Fossil fuel emissions should be used for the emissions-driven historical and future scenario simulations. C4MIP will provide gridded, annual $CO_2$ emissions from burning of fossil fuels, from the beginning of 1850 to the end of 2014 for the historical simulation and through to the end of 2100 for ssp5-8.5. They will be provided on land-points on a $1° \times 1°$ grid. It is up to model groups to re-grid or interpolate these emissions to suit their own model. Global annual totals must be conserved and must match the global annual totals of the gridded data provided. Conserving the global annual total is more important than the spatial patterns of emissions.

- C4MIP strongly recommends that land-use carbon emissions are simulated internally by applying the land-use forcing by Hurtt et al. (2016). In the event that this is not possible in a model C4MIP will provide annual land-use carbon emissions mainly based on the results of two bookkeeping models: BLUE (Hansis et al., 2015) and Houghton (Houghton et al., 2012). For the years 1850 to 2010 the average result of these two bookkeeping models defines the global emission rate, whereas the spatial distribution of the emissions is taken solely from BLUE at a 0.5 degree resolution. This approach provides input emissions more spatially consistent with the land-use forcing

applied to models than population-weighted spatial patterns used in CMIP5. For the years 2010 to 2014 the global land-use emission rate is specified by the Global Carbon Project (Le Quéré et al., 2015) and the spatial pattern is that of BLUE at the year 2010. At the time of writing this C4MIP protocol, future land-use scenarios have not yet been processed within LUH2. Our intention is that for the future scenarios we will provide gridded land use emissions using global totals from the scenario and the spatial pattern either provided from the scenario or from the BLUE spatial pattern for 2010. As with fossil fuel emissions it is up to model groups to re-grid or interpolate these emissions to suit their own model. Global annual totals must be conserved and must match the global annual totals of the gridded data provided.

### 3.3.2 Land-use and land-use-induced land cover change

LULCC affects climate via two aspects in CMIP6 simulations. In both concentration-driven and emission-driven simulations LULCC alters the distribution of vegetation covering the land surface, with consequences for the exchange of heat, water, and momentum with the atmosphere. Its effects on terrestrial carbon stocks allow us to infer LULCC emissions, more accurately labelled the "net LULCC flux" (Brovkin et al., 2013). In emission-driven simulations the net LULCC flux influences the atmospheric $CO_2$ concentration, contributing to subsequent carbon cycle feedbacks (e.g., Strassmann et al., 2008; Arora and Boer, 2010; Pongratz et al., 2014).

The LULCC forcing for the historical simulations will be based on the protocol and forcing data provided by CMIP6 for the DECK and the historical CMIP6 simulations. LULCC is kept fixed at its pre-industrial state for all 1pctCO2 simulations (fully coupled, biogeochemically and radiatively coupled versions). It is essential that the biogeochemically coupled simulations required for C4MIP of the historical and future SSP simulations and their extensions to 2300 use identically the same LULCC forcing as for the parallel ScenarioMIP simulations.

### 3.3.3 N-deposition

Models including a nitrogen cycle are encouraged to use a consistent set of forcings of anthropogenic nitrogen deposition as drivers for the respective ocean and land biogeochemical components. Rates of speciated nitrogen deposition at the land and ocean surface are not available from observations and so need to be determined by models. C4MIP will coordinate with CCMI to provide gridded, time varying fields of nitrogen deposition from chemistry transport models (CTMs) for use as driving inputs in C4MIP simulations (http://www.met.reading.ac.uk/ccmi/?page_id=375). This will be provided partitioned into four categories of wet or dry and oxidized or reduced N deposition velocities at the bottom of the atmosphere. If a model requires more or fewer categories or species of nitrogen deposition then it is up to the model group to produce these. When aggregating or disaggregating components of deposition the total amount of reactive nitrogen should be conserved. Inputs into the land biosphere depend on vegetation characteristics, and these aspects should be dealt with by the individual model groups.

C4MIP simulations should use N deposition fields as follows:

- Pre-industrial control (piControl and esm-piControl) should use time-invariant, but spatially explicit, N deposition appropriate to 1850. This is so that there are no discontinuities in carbon pools or fluxes at the beginning of the historical simulation.

- Historical (historical, esm-hist, hist-bgc) and future scenarios (esm-ssp585, ssp585-bgc, ssp585-bgcExt, ssp534-over-bgc, ssp534-over-bgcExt) should use the provided time-varying N deposition data derived from CTM simulations. It is essential that all C4MIP simulations use identically the same N deposition fields for the C4MIP simulations as the parallel DECK, Historical and ScenarioMIP simulations.

- The idealised simulations (1pctCO2, 1pctCO2-bgc, 1pctCO2-rad) should also use the time-invariant pre-industrial N deposition as used in the control runs, as $CO_2$ is the only time varying forcing in these experiments.

- For the first time, C4MIP requests additional idealised simulations (1pctCO2Ndep, 1pctCO2Ndep-bgc) designed to quantify the effect of N deposition on the carbon-climate and carbon-concentration interactions. These simulations should use an idealised scenario of time-varying N deposition as follows. A scenario will be generated by adding to the pre-industrial base-line the geographically explicit difference between the year 2100 SSP5-8.5 N deposition scenario and pre-industrial values, such that the relative growth rates of N deposition and $CO_2$ match and the global total N deposition at the time when atmospheric $CO_2$ concentrations reach the SSP5-8.5 value for the year 2100 correspond to the year 2100 N deposition total. C4MIP will generate these fields of N deposition and make them available as annual fields to be applied in these idealised simulations.

If the ESM simulates atmospheric chemistry and composition and therefore provides N-deposition internally, then this can be used in place of a prescribed field of N-deposition for the control, historical and scenario simulations. However, irrespective of whether an ESM generates N deposition or not, for the 1% idealised simulations, it is preferable to use the provided fields as anomalies which should be added to the ESM's pre-industrial N deposition fields.

The provided N-deposition data will cover both land and ocean, but we acknowledge that some models have their own established sources of reactive nitrogen to the oceans and to change this would require costly repeat-spinup simulations. So it is left to the model groups' discretion how to apply N-deposition to the ocean. If a source other than provided by C4MIP is used this should be documented and made available to aid analysis.

**3.3.4 Carbon isotopes**

Models including carbon isotopes ($\delta^{13}C$ and $\Delta^{14}C$) in land or ocean realms are encouraged to simulate and report variables relating to carbon isotopes for control, historical and future scenario simulations.

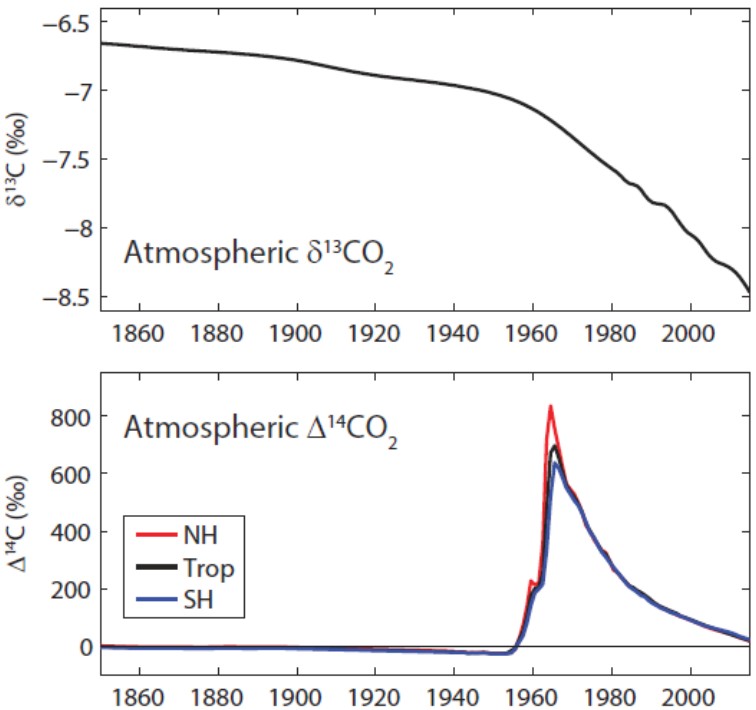

*Figure 4. Carbon isotopes in atmospheric CO$_2$ for the historical period 1850-2014. Data for $\delta^{13}$C is from Law Dome and South Pole (Rubino et al., 2013) and Mauna Loa (Keeling et al., 2001) and includes smoothing of the observations. Data for $\Delta^{14}$C is compiled from Levin et al. (2010) and other sources (I. Levin personal communication), following a similar dataset used by Orr et al. (2000).*

For historical concentration-driven runs (piControl, historical and hist-bgc), atmospheric $\delta^{13}$CO$_2$ and $\Delta^{14}$CO$_2$ forcing based on observations will be provided (figure 4). The atmospheric forcing datasets will be available at the C4MIP website. We also plan to make available atmospheric forcing data for carbon isotopes for the ssp585 scenario and for other scenarios and extensions using a simple carbon cycle model, following Graven (2015).

Carbon isotopes are only requested to be simulated in land and ocean model components using the provided historical or future atmospheric forcing datasets for $\delta^{13}$CO$_2$ and $\Delta^{14}$CO$_2$. It is not requested that atmospheric $\delta^{13}$CO$_2$ and $\Delta^{14}$CO$_2$ be simulated by ESMs, even for emission-driven simulations of atmospheric CO$_2$.

### 3.3.5 Other forcings

If the model requires any other external forcing not documented here, for example deposition of phosphorous, then it is at the model groups' discretion how to provide it. In the case of a model with an interactive phosphorous cycle we recommend the

forcing data is prepared in a way analogous to the nitrogen deposition described above. We recommend modelling groups to contact C4MIP for more details if this is applicable. Any additional forcings must be documented through the CMIP metadata process or in the appropriate model description paper.

## 4. Output requirements

It is vital for accurate analysis and model intercomparison that every model adheres to the definitions of each output variable in order for like-for-like comparison to be made. In this section we describe in detail each requested output variable. The data request will be documented separately (by the WGCM Infrastructure Panel; https://www.earthsystemcog.org/projects/wip/) and will list the required variables output for each CMIP6 simulation along with their precise variable names, description and required units. Here we aim to describe each variable so that its implementation and use are made consistent across all models and analyses.

### 4.1 Land

### 4.1.1 Land carbon cycle variables

The primary aim of C4MIP is to compare the aspects of the global carbon cycle and its response to environmental changes across the participating ESMs. To achieve this objective, it is essential that all carbon stocks and fluxes are reported so that total amount of carbon in the system can be tracked and their conservation checked. To achieve this, compulsory tier-1 diagnostics have been defined that as simply as possible close the carbon cycle. Desirable tier-2 diagnostics should also be reported where possible which allow more detailed analysis by breaking down tier-1 output into sub-components.

### *Land carbon pools: tier-1*

Figure 5 shows the requested carbon cycle stores over land. Tier-1 variables are intended to be simple but still capture the total land carbon store. Tier-2 variables provide the same information as the tier-1 variables but in more detail. As shown in Figure 5 the total carbon can be calculated from tier-1 variables and is not the combined sum of tier-1 and tier-2 variables.

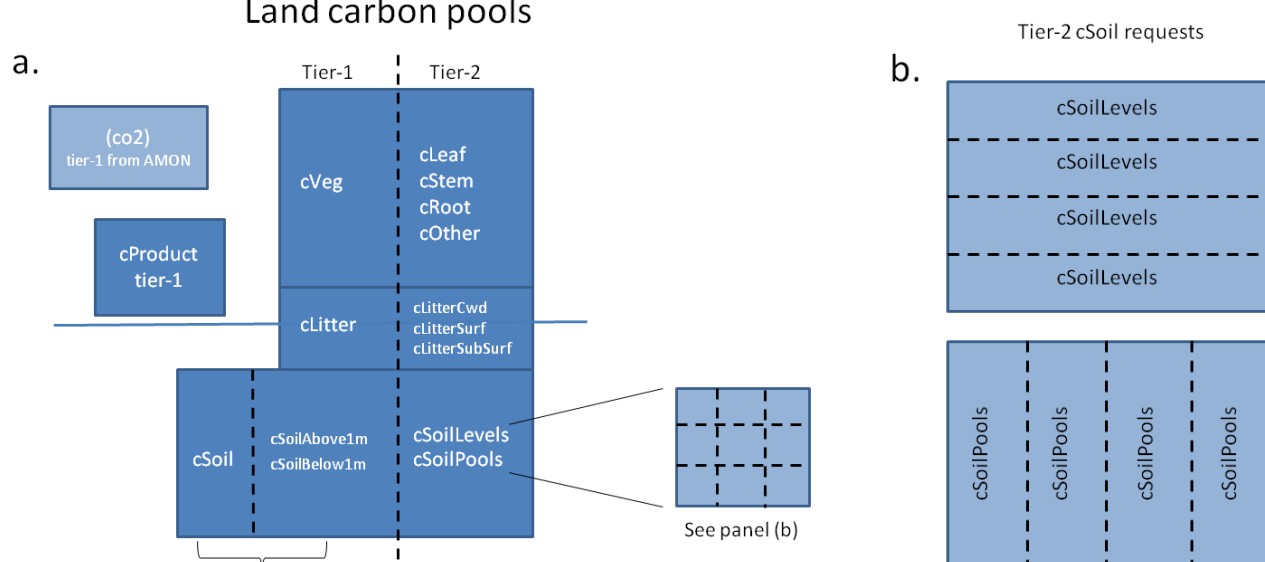

Figure *5: (a) Requested tier-1 and tier-2 variables representing land carbon pools. Although not a land carbon quantity, atmospheric $CO_2$ is shown here for completeness.(b) Detailed view of the tier-2 breakdown for soil carbon by vertical level (cSoilLevels) and by soil carbon pool (cSoilPools).*

The carbon stored in the vegetation-litter-soil system is simply represented by tier-1 variables, cVeg, cLitter and, cSoil respectively. For models which do not represent a vertical discretization of soil carbon, all soil carbon should be reported simply as cSoil. Additionally in teir-1 for models with vertically discretized soil carbon we request output on the vertical distribution above and below 1m depth (cSoilAbove1m, cSoilBelow1m). These should be reported in addition to cSoil, such that cSoil=cSoilAbove1m+cSoilBelow1m. The rationale for requesting this is the availability of several observation-based datasets that report soil organic matter content to 1 m depth. It is important that any evaluation of cSoil outputs against observed datasets makes use of the appropriate depth of soil in both the observations and model outputs.

A fourth pool, cProduct, represents the carbon stored in product pools (such as harvested wood, paper products, and furniture etc) as a result of anthropogenic land-use change. The total carbon stored per unit area on land is then simply:

$$cLand = cVeg + cLitter + cSoil + cProduct \qquad (1)$$

Some models may not explicitly simulate a litter pool distinct from their soil carbon pool. In this case cLitter should be reported as zero. We would normally expect cProduct to be non-zero in simulations which include anthropogenic land-use or land-use change. Hence, for the idealised 1% per year increasing $CO_2$ simulations (biogeochemically-, radiatively or fully-

coupled) we would expect models to report cProduct=0. For models whose land-use fluxes contribute straight to the atmosphere and/or to their litter or soil carbon pools, but not to the product pools, cProduct=0 should also be reported for historical and scenario simulations. Obviously, for models which do not simulate the effect of LULCC on the carbon cycle, cProduct will also be expected to be zero.

### *Land carbon pools: tier-2 vegetation and litter carbon*

Tier-2 output variables allow for more detailed breakdown and analysis of their parent carbon stores. They are sub-components of their parent tier-1 variables, and not additional stores. For example, the vegetation carbon pool can be represented by carbon in the leaf, stem and root and possibly other (e.g. fruit) components. For models which report these

tier-2 variables, the total amount of carbon per unit area should be identical to the tier-1 variable, i.e.:

$$cVeg \;=\; cLeaf \;+\; cStem \;+\; cRoot \;+\; cOther \tag{2}$$

The same applies for the litter carbon pool, which is requested to be broken down into coarse woody debris (cLitterCWD) and above- and below-surface litter (cLitterSurf, cLitterSubSurf) pools. Where a model has a continuous profile of litter with

depth, take above and below 10cm as the definition of above and below the surface. CWD here is assumed to be on the surface.

### *Land carbon pools: tier-2 soil carbon*

For CMIP5 the soil carbon pool was requested to be divided into components with fast, medium and slow turnover

timescales. However, this distinction was not found useful by the community and as a result was not used in many analyses. For CMIP6, we request a breakdown in two different ways (figure 5b). Firstly models with a vertical structure to their soil carbon are requested to report total soil carbon for each soil layer. In the same way as soil moisture or temperature, this should be reported as a multi-level output, cSoilLevels. As the structure for this may vary between models it is essential that the model is thoroughly documented. The sum of soil carbon over all cSoilLevels should be identical to the total cSoil tier-1

variable.

Most soil carbon models represent multiple soil carbon pools (such as fast or slow turnover, or decomposable and resistant organic material). In order to be able to diagnose and evaluate the turnover rates of carbon within the terrestrial system we make a second tier 2 request to report individual soil carbon pools (figure 5b, lower panel). It is also required to report the

turnover rate (tSoilPools: defined as 1/residence time) for each pool. The pool-flux structure of each model should be fully documented in its model description paper. This output will enable reduced complexity approaches (e.g. Xia et al., 2013) to recreate and analyse the soil carbon dynamics within each model. The sum of soil carbon over all cSoilPools should be identical to the total cSoil tier-1 variable.

### *Land carbon pools: tier-2 carbon on sub-grid tiles*

A final tier-2 breakdown is required to report the main stores and fluxes separately for different land cover types. The LUMIP data request (Lawrence et al., 2016) requests carbon pools and fluxes for four land cover types: crop, pasture, primary and secondary land (combined as one tile), and urban. For C4MIP we additionally request a breakdown of carbon pools and fluxes within "primary and secondary" land onto tree, shrub and grass separately. Section 4.1.4 describes the C4MIP requested output of land cover fractions. Carbon pools (cVeg, cLitter and cSoil) and fluxes (gpp, npp, ra, rh) are therefore requested on the treeFrac, shrubFrac, grassFrac, cropFrac and pastureFrac fractions shown in figure 11. Table 3 lists all of these requests.

| portion of gridbox | pools | Fluxes |
|---|---|---|
| treeFrac | cVegTree, cLitterTree, cSoilTree | gppTree, nppTree, raTree, rhTree |
| shrubFrac | cVegShrub, cLitterShrub, cSoilShrub | gppShrub, nppShrub, raShrub, rhShrub |
| grassFrac | cVegGrass, cLitterGrass, cSoilGrass | gppGrass, nppGrass, raGrass, rhGrass |
| cropFrac | cVegCrop, cLitterCrop, cSoilCrop | gppCrop, nppCrop, raCrop, rhCrop |
| pastureFrac | cVegPast, cLitterPast, cSoilPast | gppPast, nppPast, raPast, rhPast |

*Table 3. Summary of tier-2 data request of carbon pools and fluxes by sub-grid land cover fraction.*

### *Land carbon fluxes*

Equally important to the land carbon pools are the fluxes going into and out of them which will allow us to gain insight into how the pools have changed and why. For ease of understanding we have adopted a convention for newly defined variables that a carbon pool is prefixed by a "c" (as in cVeg or cSoil) and a flux by an "f" (as in fLandToOcean). Some existing variables (e.g. gpp and npp) do not conform to this but are considered to be well known and do not need to be changed.

Figure 6 shows the variables requested for terrestrial carbon fluxes. Similar to land carbon pools, the objective of tier-1 fluxes is to capture the primary system behaviour, and tier-2 fluxes provide breakdown within the tier-1 fluxes allowing for a more detailed analysis. The directions of the arrows indicate the sign-convention of the flux which is considered positive in the direction in which the arrows are pointing. For example, gross primary productivity (gpp) is positive downwards indicating flux of carbon *from* the atmosphere *to* the vegetation, whereas autotrophic respiration (ra) is positive upwards indicating flux of carbon *from* the vegetation *to* the atmosphere.

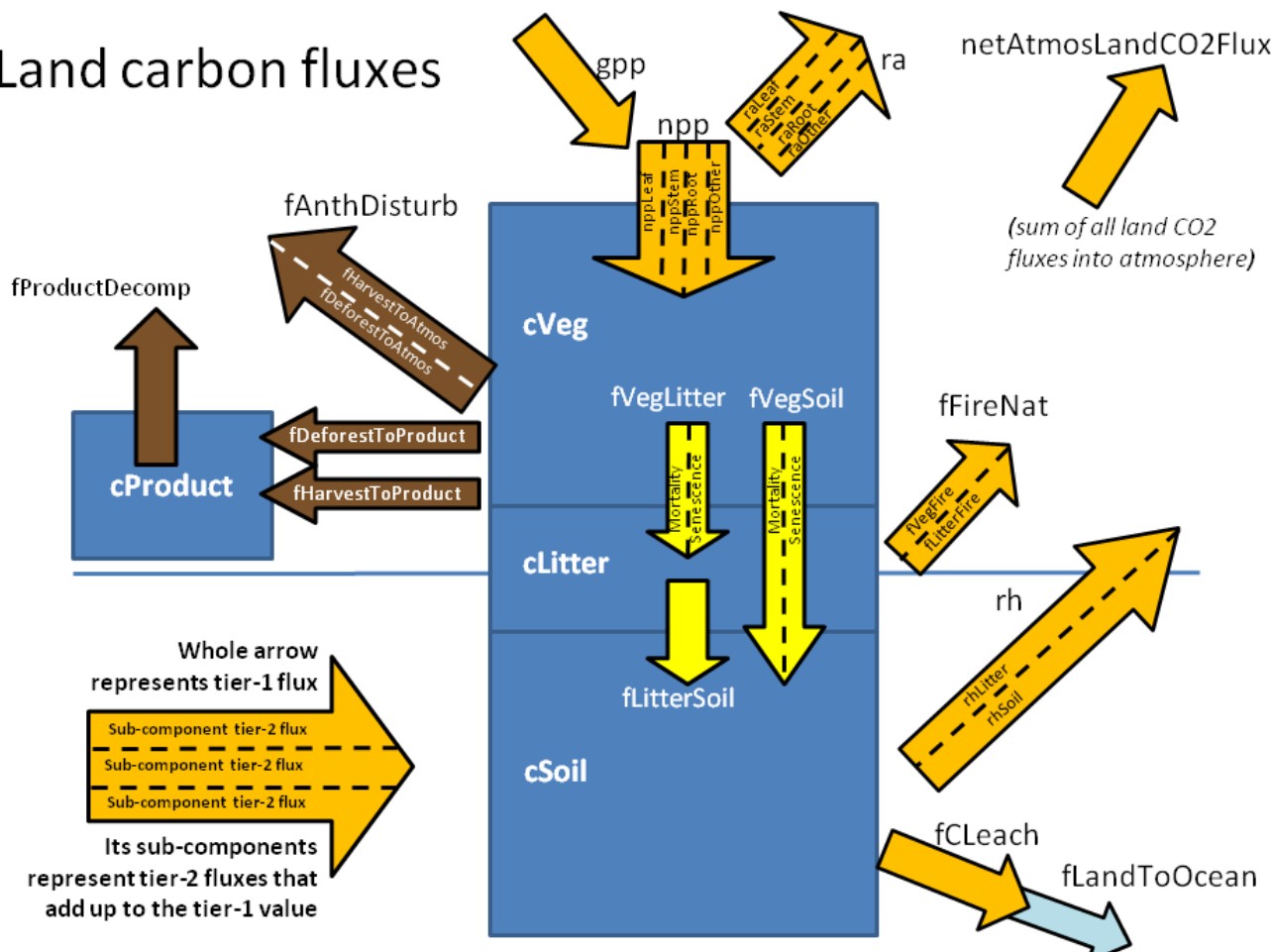

*Figure 6: Requested tier-1 and tier-2 variables representing land carbon fluxes. The colours of the arrows correspond to the type of flux. The orange arrows represent "natural" fluxes that represent pathways of carbon exchange between the land and atmosphere. These natural fluxes would generally be expected to be non-zero in all simulations. The brown arrows represent fluxes associated with anthropogenic disturbance between land pools or between the land and the atmosphere. These fluxes would be expected to be non-zero in simulations that implement anthropogenic land use change based on land-use change scenarios. The yellow arrows represent internal fluxes within the veg-litter-soil system. Finally the blue arrow represents carbon loss from land to the ocean, which may be a subset of leached carbon, although not all models may simulate this flux.*

Gross primary productivity, gpp, is the flux of carbon from the atmosphere to the vegetation that is associated with photosynthesis. Net primary productivity (npp) represents the carbon uptake by vegetation after the autotrophic respiration (ra) costs have been taken into account (npp = gpp - ra). Both ra and npp are sub-divided into tier-2 outputs representing flux

from the leaf, stem and root, components, respectively plus also a category "other" which should include all the components (if any) reported under cOther tier-2 carbon pool. Also, similar to land surface pools, the sum of the tier-2 fluxes must be identical to their parent tier-1 flux.

$npp = nppLeaf + nppStem + nppRoot + nppOther$                                                      (3)

Heterotrophic respiratory flux (rh) and $CO_2$ emissions associated with natural wildfires (fFireNat) represent carbon loss from the land carbon stores to the atmosphere. rh is requested to be sub-divided into its tier-2 components from the litter and soil pools. Similarly, fFireNat is sub-divided into fire $CO_2$ emissions from vegetation and litter carbon pools. Note, that fFireNat
should not include $CO_2$ emissions from fires associated with anthropogenic land use change.

Anthropogenic land-use change or land management can result in transfer of carbon out of the vegetation, litter and soil carbon pools either directly to the atmosphere (fAnthDisturb) or to the product pool. fAnthDisturb is proposed to be split into fluxes due to land-cover change (fDeforestToAtmos) or management (fHarvestToAtmos), if this distinction is made in the
model. Anthropogenic fires, associated with LUC, should be included in fAnthDisturb. Fluxes into the product pool should similarly be reported as either fDeforestToProduct or fHarvestToProduct. Decomposition of carbon in the product pool represents a carbon flux back to the atmosphere (fProductDecomp).

Due to the complexity of the processes involved, especially in the treatment of land-use and management, and the growing
complexity in the manner in which LUC is represented in the models, it is possible that this simple framework may not be completely compatible with all models. It is simply not possible to define in advance of CMIP6 a framework that may cover every possible flux in every model. Our request is, therefore, that all fluxes of carbon are reported somewhere, in the best possible way that they may fit within the framework shown in Figure 6, and not missed. This will ensure conservation of carbon within the reported variables.

An example of differences in model structure and processes is the manner in which litter from the vegetation pool is transferred to the soil carbon pool. Some models simulate litter fall from vegetation into the litter pool and then subsequent assimilation into the soil carbon pool. Some models may also simulate this flux directly from vegetation to soil carbon. In either case tier-2 breakdown of the litterfall flux due to senescence (normal turnover) and mortality is requested; this
breakdown is expected to help to diagnose changes in turnover time of the litter and soil carbon pools.

Figure 6 also forms the basis of carbon conservation properties that must be obeyed by the reported outputs. These include the manner in which fluxes should add up and that the rate of change of carbon in carbon pools must be equal to the sum of

fluxes going in and out of the pools, or equivalently changes in pools must be equal to the sum of time integral of the fluxes into and out of the pools.

$$gpp = npp + ra \tag{4}$$

$$\frac{d\,cVeg}{dt} = npp - fVegLitter - fVegSoil - fAnthDisturb - fDeforestToProduct - fHarvestToProduct - fVegFire \tag{5}$$

$$\frac{d\,cLitter}{dt} = fVegLitter - fLitterSoil - fLitterFire - rhLitter \tag{6}$$

$$\frac{d\,cSoil}{dt} = fLitterSoil + fVegSoil - rhSoil - fLandToOcean \tag{7}$$

$$\frac{d\,cProduct}{dt} = fDeforestToProduct + fHarvestToProduct - fProductDecomp \tag{8}$$

We define a new variable, *netAtmosLandCO2Flux*, which is the total flux of $CO_2$ from the land to the atmosphere. It should encompass every flux from land to atmosphere so that the total from each model can be compared without having to know model details of which component fluxes to sum. Due to differences in naming convention we have chosen not to call this

NBP (net biome productivity). This is an essential tier-1 variable requested from all C4MIP simulations.

**4.1.2 Land nitrogen cycle variables**

Figures 7 and 8 summarize the requested terrestrial nitrogen pools and flux variables from models that include a representation of terrestrial nitrogen cycle and its coupling to the terrestrial carbon cycle. The nitrogen pools are designed to parallel their corresponding carbon stores as closely as possible, giving primarily the storage of nitrogen in the vegetation

(nVeg), litter (nLitter) and soil organic matter (nSoil) pools. Additionally, we are requesting mineral nitrogen in soil (nMineral), which is sub-divided into tier-2 variables representing ammonium (nMineralNH4) and nitrate (nMineralNO3) mineral nitrogen. We don't envisage much interest in the nProduct variable (nitrogen stored in anthropogenic product pools), but it is required as a tier-1 output in order to close the nitrogen budget and ensure mass conservation of analyses. There will also be likely little interest in separating nLitter into its tier-2 components nLitterCwd, nLitterSurf and nLitterSubSurf but

these variables are being requested for consistency with their carbon counterparts.

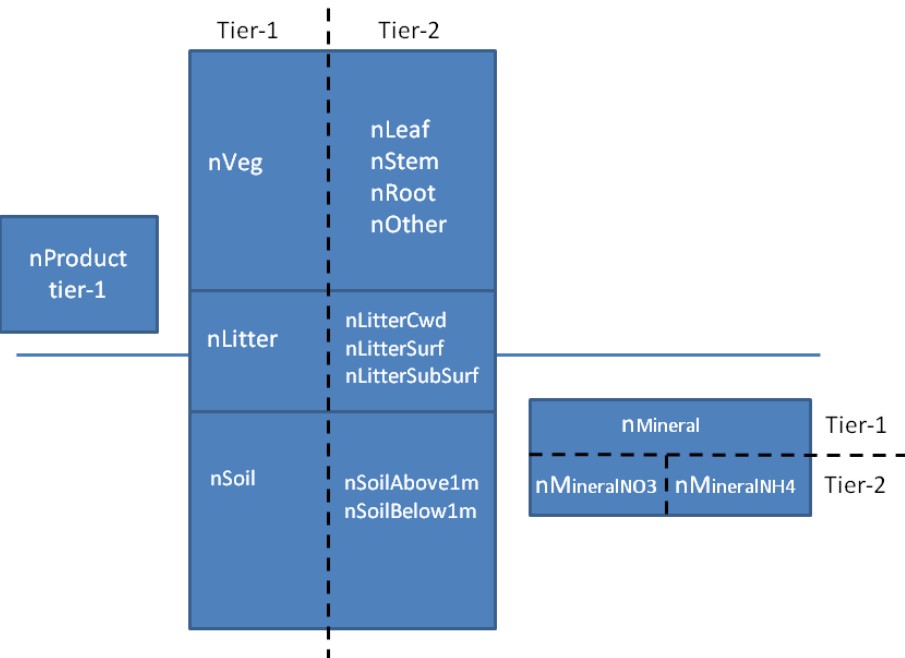

*Figure 7: Requested tier-1 and tier-2 variables representing land nitrogen pools.*

Requested fluxes associated with the flow of nitrogen over land are summarized in Figure 8 and differ more from their carbon counterparts than do the carbon and nitrogen pools. As with the pools, all fluxes should be reported somewhere in order to be able to close nitrogen cycle budget over land and ensure mass conservation of analyses. As with carbon fluxes, the sign convention of the flux is considered positive in the direction in which the arrows are pointing

Nitrogen enters the terrestrial ecosystems either through anthropogenic inputs (which can be either atmospheric deposition, fNdep, or fertiliser input fNfert) or through biological fixation (fBNF). Flows between vegetation, litter and soil organic N pools mirror the carbon fluxes, but with additional terms that represent inorganic mineral nitrogen uptake by vegetation (fNup) and the net mineralisation flux, i.e. the difference between gross mineralisation and immobilisation, from the dead litter and soil organic matter pools to the mineral nitrogen pool (fNnetmin). fNnetmin should be reported as positive *into* the nMineral pool. Negative values of fNnetmin then imply net immobilization.

The tier-1 variables that represent the loss of nitrogen from the primary terrestrial pools of vegetation, litter and soil organic matter include fluxes due to anthropogenic disturbance: either into the LUC product pool (fNproduct) or loss direct to the atmosphere fNAnthDisturb and loss from the mineral nitrogen pool (fNloss). In order to conserve nitrogen, all losses of N must be reported into one of these variables. fNloss may be further sub-divided (if represented in the model) into tier-2 outputs of gaseous loss to the atmosphere (fNgas) and loss of dissolved organic and inorganic nitrogen through leaching (fNleach) i.e. fNloss = fNgas + fNleach. If represented in the model, fNgas can be split into that due to fire and non-fire. A further breakdown of tier-2 fluxes is also requested, if available, but these do not necessarily have to add up to the tier-1 flux value: fNOx and fN2O are components (but do not necessarily have to add up to fNgas) and may be of interest for evaluation activities or coupling to atmospheric chemistry models. fNLandToOcean may be a subset of fNleach and is of interest for studying the impact of terrestrial nitrogen cycle on coastal ocean ecosystems.

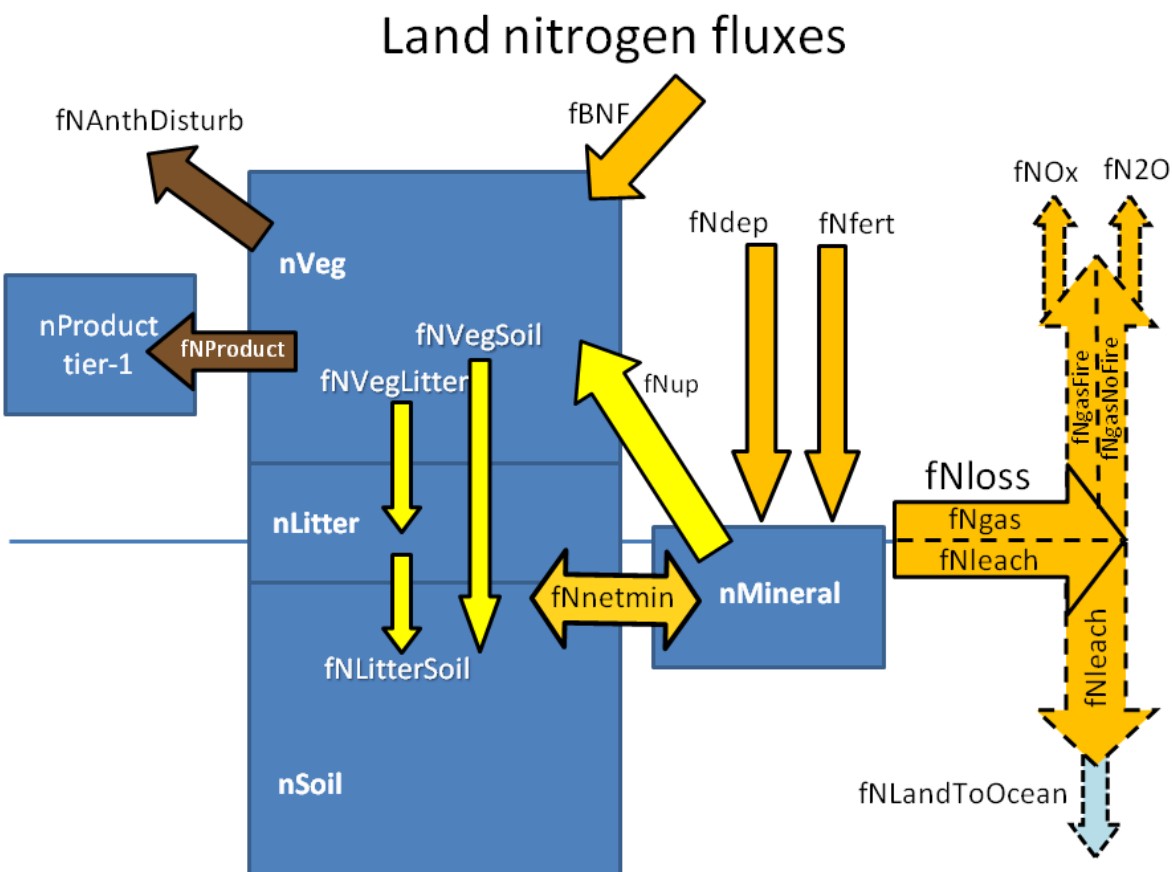

*Figure 8: Requested tier-1 and tier-2 variables representing land nitrogen fluxes.*

**4.1.3 Land physical variables**

While most variables representing the land surface physical state and water fluxes will likely be requested by the land surface, snow and soil moisture model intercomparison project (LS3MIP, van den Hurk et al., 2016) and land use model intercomparison project (LUMIP, Lawrence et al., 2016), C4MIP is requesting some basic land surface physical variables as well. These include soil moisture and temperature, vegetation leaf area index (LAI) and height, and basic water fluxes.

*Physical state variables*

Figure 9 shows the state variables requested that characterize the physical vegetation structure (through leaf area index and vegetation height) and the physical state of the soil (through the soil moisture and temperature of a model's soil layers).

The only tier-1 state variable requested for vegetation structure is leaf area index (LAI), which represents the area of leaves per unit area of ground. Vegetation height may also be considered an important evaluation metric but this is requested as tier-2 variable. It is likely more useful to distinguish vegetation height by vegetation type, i.e. by tree, shrub, grass and crop. If this distinction is not made or unavailable in a model then only the grid-averaged vegetation height may be reported.

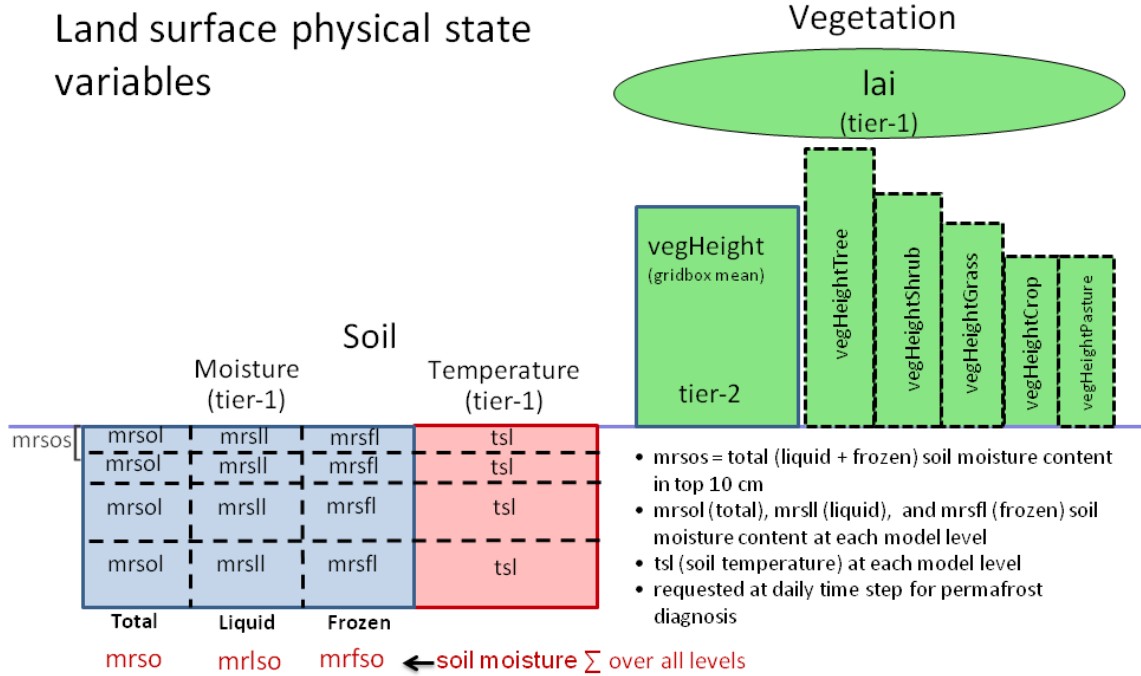

*Figure 9: Requested state variables that characterize the physical vegetation structure and the physical state of the soil.*

Soil moisture and temperature are requested as tier-1 variables to be able to analyze carbon and moisture fluxes together and to identify the role of the physical state of the soil conditions on carbon stores and fluxes. The total, liquid and frozen soil moisture contents are aggregated and disaggregated in various ways as shown in Figure 9 and described below:

- soil temperature, tsl, is requested for each model level
- soil moisture is requested as:
- total soil moisture content (sum of frozen and liquid) in the top 10cm, mrsos,
- total (mrsol), liquid (mrsll) and frozen (mrsfl) soil moisture content at each model level, and
- column integrated total (mrso), liquid (mrlso) and frozen (mrfso) soil moisture contents
- Additionally, a total water diagnostic, mrtws, is requested as tier-2 variable. This includes all soil moisture as reported above (mrso) but additionally includes water from other stores such as sub-grid lakes, aquifers, or rivers if they are represented in the model.

*Physical water fluxes*

Figure 10 summarizes the small number of land surface hydrological fluxes being requested. As with the carbon and nitrogen fluxes the sign convention is shown by the direction of the arrows.

- prveg represents precipitation intercepted by the canopy, and evspsblveg represents evaporation from the canopy leaves (including sublimation)
- evspsblsoi represent evaporation from bare soil, and includes sublimation
- tran represents transpiration flux of moisture through the vegetation and out of the leaf stomata
- Models may represent runoff in multiple ways. The runoff variables requested here are distinct from river/stream flow variables which other MIPs may request. Runoff is represented in depth units (kg m$^{-2}$ s$^{-1}$), while river/stream flow represents volume of water per unit time generated by integrating runoff from upstream grid cells (m$^{3}$ s$^{-1}$). mrros represents the surface runoff from each grid cell, and mrro represents the total runoff (including from the surface, the subsurface and any drainage through the base of the soil model)

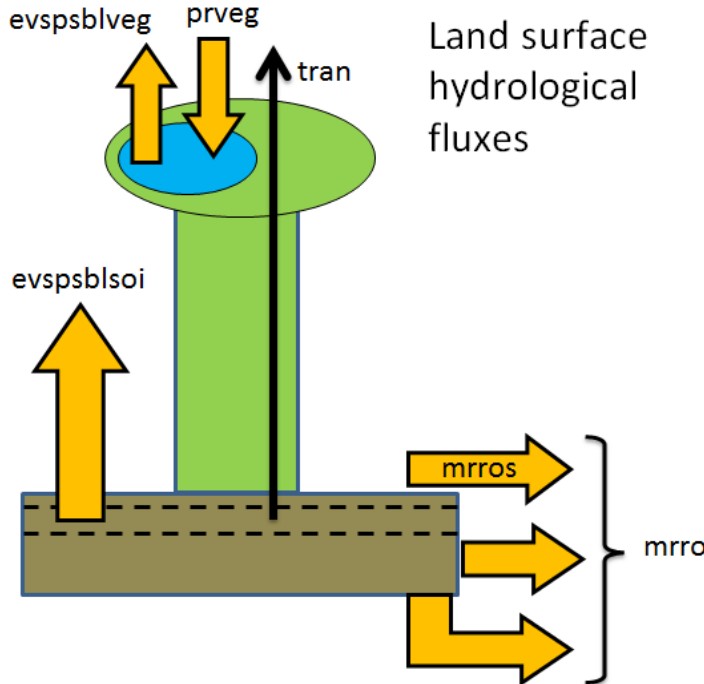

*Figure 10: Requested land surface hydrological flux variables.*

### 4.1.4 Land cover state variables

5  Figure 11 summarizes the land cover variables requested from all models. As with other requested variables, these are categorised as simpler tier-1 variables which represent the primary land cover types, while the tier-2 variables further break down the tier-1 variables into more detail. Tier-1 land cover variables are required from all models so that the land-cover is completely described. Where possible modelling groups are requested to provide the additional details through tier-2 variables. It is important that the combined totals of tier-2 variables agree with their tier-1 counterparts.

A grid cell is described in terms of vegetation fractional coverage (vegFrac), fractional coverage of bare soil (baresoilFrac) and a residual term (residualFrac) that may include fractional coverage of urban areas, sub-grid scale lakes and stony outcrops. For grid cells at the continental edges, a fraction of the grid cell may also be covered by open ocean/sea. The vegFrac is further subdivided into fractional coverage by trees (treeFrac), shrubs (shrubFrac), grasses (grassFrac) crops

15  (cropFrac) and pasture (pastureFrac). Crop and Pasture fractions are the same as requested by LUMIP (Lawrence et al., 2016). Tree, shrub and grass fractions represent additional detail within the LUMIP tile called "primary and secondary land". All land cover must be reported, such that:

$$VegFrac + baresoilFrac + residualFrac + SeaFrac = 1 \tag{9}$$

$$treeFrac + shrubFrac + grassFrac + cropFrac + pastureFrac = VegFrac \qquad (10)$$

The tier-2 land cover variables follow the separation of trees based on their leaf structure (broadleaf and needleleaf) and leaf phenology (evergreen and deciduous) as treeFracNdlEvg, treeFracNdlDcd, treeFracBdlEvg, treeFracBdlDcd. The fractional coverage of grasses, crops and pasture is separated into $C_3$ and $C_4$ variants based on their photosynthetic pathway. Tier-2 totals should sum to be identical to their tier-1 counterparts. For example:

$$treeFracNdlEvg + treeFracNdlDcd + treeFracBdlEvg + treeFracBdlDcd = treeFrac \qquad (11)$$
$$grassFracC3 + grassFracC4 = grassFrac \qquad (12)$$

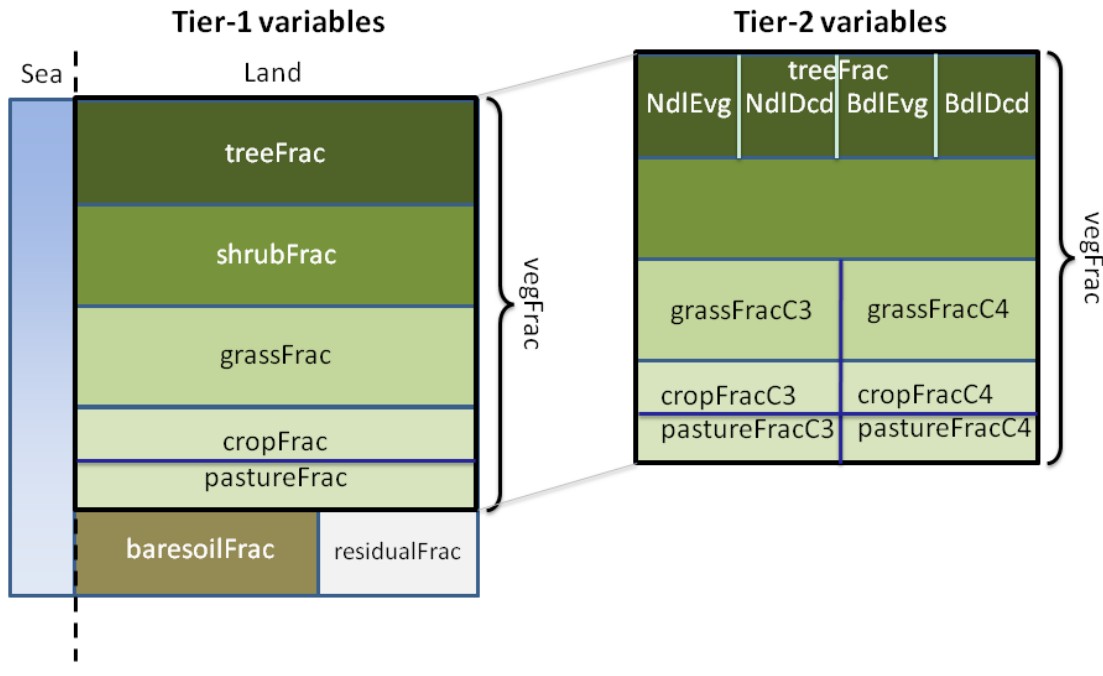

*Figure 11: Requested land cover variables. Sea fraction is assume to be fixed, so must be reported under "climatology". Fractions must sum to 1 for every grid cell (including the sea fraction). Fractions are per grid cell, not per land area.*

### 4.1.5 Auxiliary land cover fractions and fluxes

Figure 12 shows auxiliary land cover diagnostics and fluxes that may be reported. The additional land cover types are fractions of a grid cell related to a biogeochemical process that models may specifically simulate. These include burned area (burntFractionAll) and wetland fraction (wetlandFrac). burntFractionAll is expected to include burned area from all natural and anthropogenic processes (anthropogenic fires, and land use change and management related fires). wetlandFrac is expected to include natural wetlands (dynamically calculated in the model or specified) including any area of rice paddies if it is explicitly represented. Both the burnt and wetland fractions must be reported as the fraction of the grid cell and not as fraction of the land or vegetation area. Where models also estimate natural methane wetland emissions from the wetland fraction these can also be reported (wetlandCH4prod) and must include emissions from rice paddies (if represented) to make methane emissions consistent with the reported wetland fraction. If models simulate methane uptake by soils then this may be reported as wetlandCH4cons. The net land-to-atmosphere methane flux is to be reported as wetlandCH4. Models that simulate methane emissions from wetlands and/or rice paddies may explicitly simulate the depth to the water table and this may also be reported as waterDpth. Positive values of waterDpth indicate water table is below the ground surface and negative values indicate that the water table is above the ground surface.

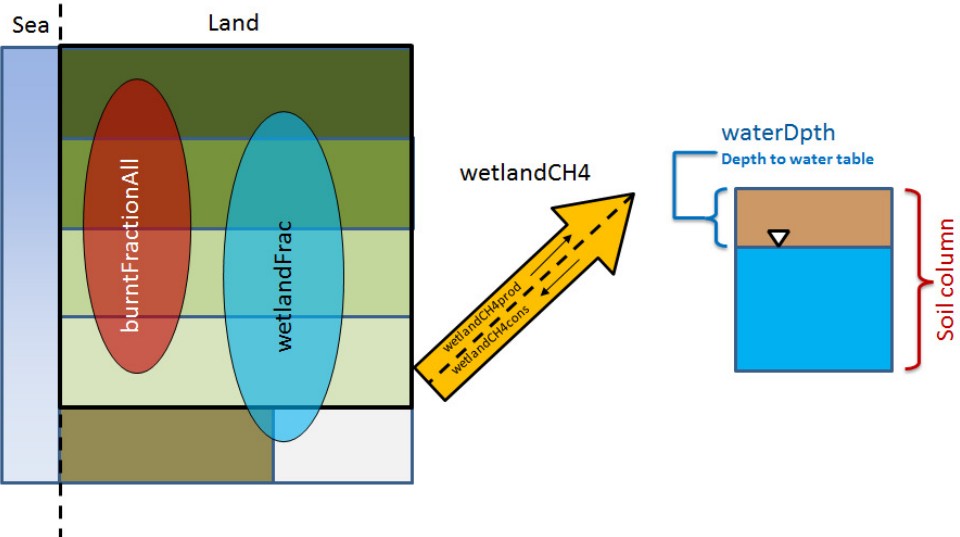

*Figure 12: Fire and wetland variables. Other than burntFractionAll, all other variables are requested as Tier 2 variables.*

### 4.2 Ocean diagnostics

Ocean biogeochemical stores and fluxes are described below. As with the land, it is important that all carbon stocks are reported so that total carbon can be tracked and conservation checked. Figures 13-16 show the requested diagnostics. Tier-1 diagnostics are intended to be simple and capture the whole ocean carbon cycle, while Tier-2 diagnostics repeat tier-1 but in

more detail. As such the total carbon is the sum of tier-1 and not the combined sum of tier-1 plus tier-2. The main (Tier-1) processes considered are: 1) gas exchange with the atmosphere that requires modelling the coupled cycle of alkalinity, and 2) biological processes coupling the carbon cycle with nitrogen, phosphorus, iron, silicon nutrients. These biological processes are centred around phytoplankton-based primary production of organic carbon, ecosystem modulation through zooplankton grazing and higher trophic interactions, sinking of organic material out of the 100 m reference level (nominal euphotic zone depth), and recycling of nutrients. Additional mechanisms working at the process level may include: biodiversity among phytoplankton, zooplankton and bacteria, dissolved organic carbon cycling, oxygen cycling and its modulation of remineralization and denitrification, $N_2$-Fixation/denitrification, flexibility in the stoichiometry among elements, sediment interactions, silicification, calcification, lithogenics, mineral ballasting of sinking material, aspects of iron cycle modulation through scavenging and the role of ligands, phytoplankton mortality by aggregation and viruses. The integral of a particular tracer XXX over model vertical levels is IntXXX, and the total time rate of change of tracer XXX is diagnosed as FddtXXX. Similarly, the time rate of change due to the sum of all biological terms acting on tracer XXX is diagnosed as FbddtXXX. XXXs is the surface value of XXX.

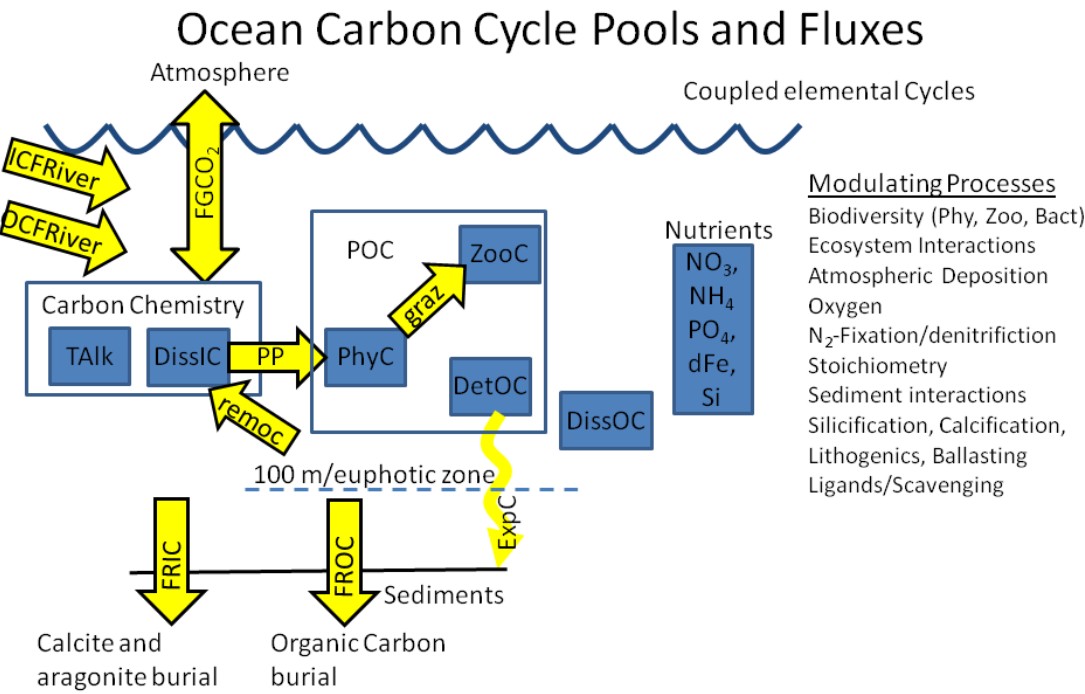

*Figure 13: Ocean Carbon Cycle Pools (blue boxes) and fluxes (yellow arrows) with associated processes. Where appropriate, pools are grouped into components like Particulate Organic Carbon (POC).*

The ocean ecosystem in ESMs typically comprises up to 5 phytoplankton functional groups: diazotrophs which can fix N2 but may take up nitrate or ammonia as well depending on the model formulation, diatoms which take up silicate to form opal tests, calcareous phytoplankton which take up dissolved carbonate and alkalinity to form calcite or aragonite tests, picophytoplankton, and miscellaneous phytoplankton in which any other phytoplankton groups are combined. Zooplankton groups may be separated by size into microzooplankton, mesozooplankton, and macrozooplankton. Combined with bacteria and detritus, these pools form the particulate organic carbon pool. Carbon stores in each of these sub-components are requested as tier-2 (figure 14) and should sum to be identical to their tier-1 counterparts.

*Figure 14: Ocean ecosystem carbon pools in terms of chlorophyll-based and carbon-based phytoplankton functional groups, zooplankton size groups, bacteria, detritus and dissolved organic carbon. as with land carbon diagnostics, the tier-2 requests are subcomponents of the tier-1 aggregate quantities. For example, ZooC should report the total carbon pool in*

*zooplankton. The sum of the tier-2 components ZooMicro, ZooMeso and ZooMisc should be identical to the tier-1 total. They are not additional pools to it.*

As shown in Figure 15, phytoplankton growth consumes dissolved organic carbon and nutrients in the presence of light to
form particulate organic carbon and oxygen through primary production (i.e. intPb), some of which is exported (i.e. expC). For each phytoplankton group, the degree of limitation by light (i.e. limIrrdiat), nitrogen (i.e. limNdiat) and iron (i.e. limFediat) availability can be diagnosed. For each elemental cycle the external sources (i.e. FSC) and removal (i.e. FRC) can be diagnosed. As model implementation of multiple factor limitation is very model dependent, limitation terms for light and nutrients should be diagnosed in a manner consistent with model implementation. For each model participant, it will be
important to document how combinations of limitation terms should be combined, multiplicatively, as the minimum, or otherwise

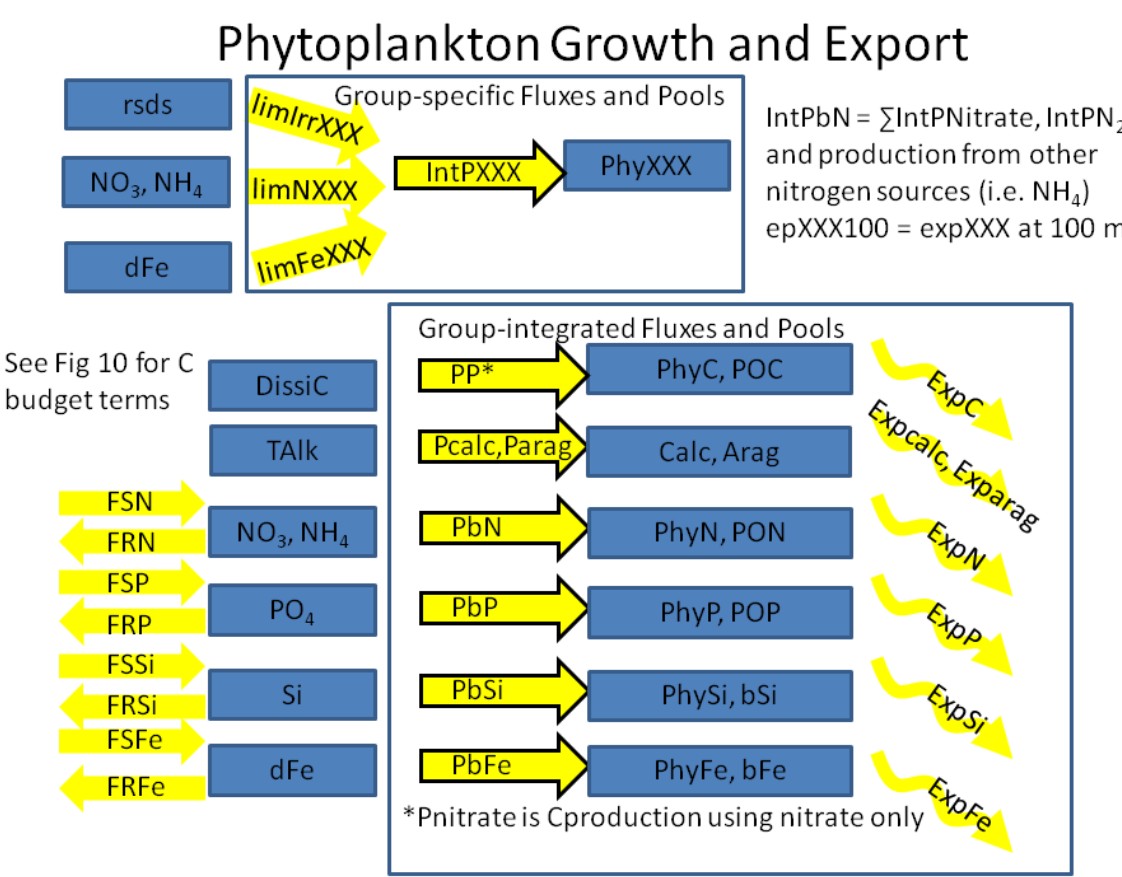

*Figure 15: Phytoplankton growth and export variables by phytoplankton group and by associated elemental cycle including
external sources and removal. Export refers to the export flux due to sinking.*

Chemistry associated with the carbon system and gas exchange is kept track of through the variables provided in Figure 16. Cycles include the full carbon system associated with dissolved inorganic carbon and alkalinity as well as additional components relevant to specific tracer analysis such as the natural carbon system that is unaffected by anthropogenic $CO_2$, and simplified abiotic dissolved inorganic carbon and abiotic alkalinity used for simulation of radiocarbon (dissic14C, dissic14Cabio).

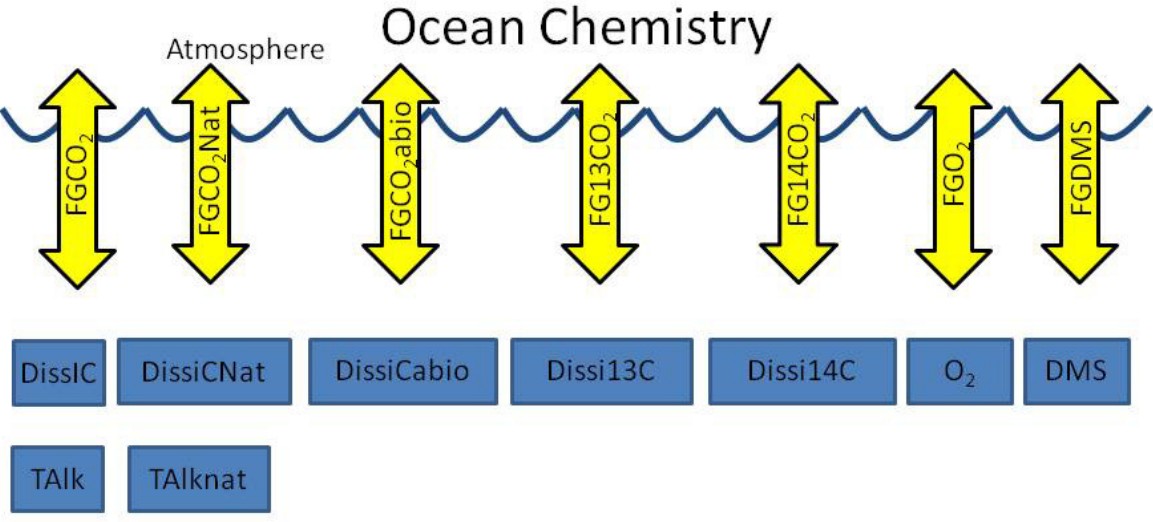

*Figure 16: Ocean chemistry including the suite of carbon system tracers and those undergoing gas exchange.*

## 4.3. Carbon isotopes

Carbon isotopes are not simulated in all models and have not been requested or used before in C4MIP analyses. For CMIP6 we request that any model that simulates isotopes of carbon (13 or 14) either on land or in the ocean report them in the same way as the tier-1 carbon outputs.

Figure 17 shows carbon isotope diagnostics which are requested. These represent stocks of carbon-13 and carbon-14 in both land and ocean reservoirs and their exchange fluxes with the atmosphere. Net air-sea fluxes of carbon-13 and carbon-14 and dissolved inorganic of carbon-13 and carbon-14 concentrations in the ocean are requested. On land, fluxes of carbon-13 and carbon-14 associated with gross primary productivity, autotrophic respiration and heterotrophic respiration, and stocks of carbon-13 and carbon-14 in vegetation, litter and soil are requested. The same units used for carbon should be used for carbon-13 and carbon-14. Stocks and fluxes of carbon-14 should be normalized with the standard $^{14}$C/C ratio, Rs, of 1.176 x $10^{-12}$ (Karlen et al., 1968). This means that reported stocks and fluxes of carbon-14 should be divided by Rs.

Decay of carbon-14 should use the currently accepted half-life of 5700±30 years. In ocean models, carbon-14 can be run as an abiotic variable (Orr et al., 2000) or integrated into marine ecosystem carbon cycling. If carbon-14 is run as an abiotic variable, abiotic dissolved inorganic carbon concentrations and abiotic carbon air-sea fluxes must also be reported. For carbon-13 in the ocean, we request only net air-sea fluxes of carbon-13 and carbon-13 in DIC. We do not request variables related to carbon-13 in phytoplankton or carbon-13 fluxes between DIC and phytoplankton, even though ocean models including carbon-13 are likely to include marine ecosystem cycling of carbon-13. More detail on implementing carbon isotopes in ocean models for CMIP6 can be found in Orr et al. (2016).

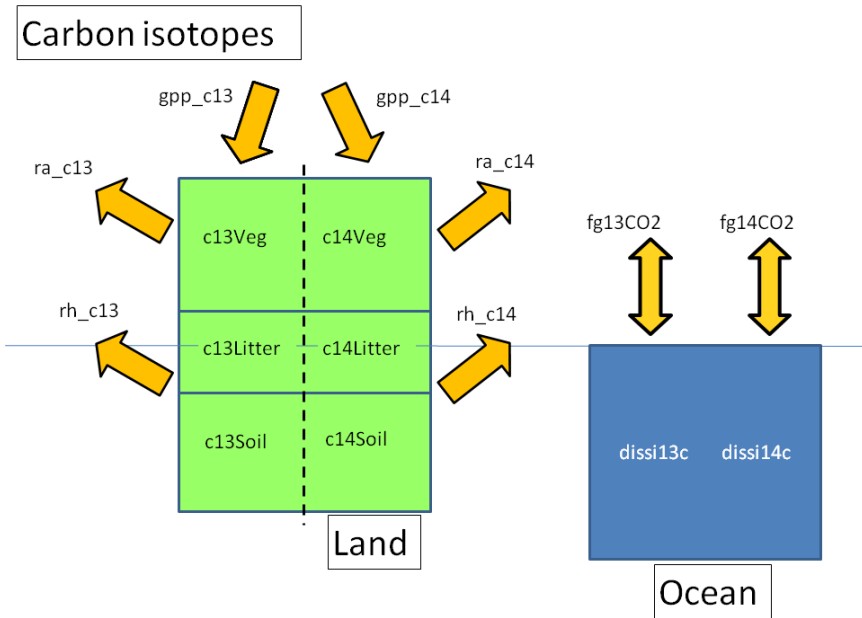

*Figure 17 Carbon isotope diagnostics. Only report for models simulating isotopes. We define:*
*c13Land=c13Veg+c13Litter+c13Soil and likewise for c14Land. As for cSoil, models with vertical discretization should also report above and below 1m separately as c13SoilAbove1m and c13SoilBelow1m and likewise for c14.*

## 5. Conclusions

Processes in the natural carbon cycle currently remove approximately half of anthropogenic emissions of $CO_2$, helping to reduce the magnitude and rate of climate change. How these processes may change in the future in response to environmental changes and direct human forcing is uncertain.

As an endorsed activity of CMIP6, C4MIP will contribute coordinated simulations and analyses targeted at 3 key carbon cycle areas. Namely:

- feedback quantification through idealised simulations. Here we hope to better understand and quantify the sensitivity of land and ocean carbon uptake to key environmental changes; and in particular the impact of climate
change on carbon uptake;
- model evaluation through analysis of historical simulations. Here we hope to build trust in projections through process-based and top-down evaluation, advancing our understanding of the strengths and weakness of ESMs and documenting progress since CMIP5;
- future projections of climate and $CO_2$ under scenarios of $CO_2$ emissions. Here we hope to better project the future
response to anthropogenic activity through $CO_2$ emissions-driven simulations which allow the full range of feedbacks to operate from CO2 emissions to the evolution of atmospheric $CO_2$ and the associated climate response.

C4MIP will focus on the coupled earth system, comprising land-atmosphere-ocean physical realms and both the terrestrial and marine carbon cycle components. Offline studies of land-only or ocean only will complement our analyses but are
outside the specific remit of C4MIP.

Over the last 2 years the C4MIP community has devised a compact and efficient set of numerical experiments to be performed with ESMs to address the above questions. In this paper we have documented the rationale and set-up of these simulations and the required outputs. This therefore constitutes the C4MIP contribution to CMIP6.

**Data availability**

As with all CMIP6-endorsed MIPs the model output from the C4MIP simulations described in this paper will be distributed through the Earth System Grid Federation (ESGF). The natural and anthropogenic forcing datasets required for the simulations will be described in separate invited contributions to this Special Issue and made available through the ESGF with version control and digital object identifiers (DOI's) assigned. Links to all forcings datasets will be made available via
the CMIP Panel website.

**Acknowledgements**

Acknowledgements. CRESCENDO project members (CDJ, PF, LB, VB, TI, SZ) acknowledge funding received from the Horizon 2020 European Union's Framework Programme for Research and Innovation under Grant Agreement No 641816. CDJ was supported by the Joint UK DECC/Defra Met Office Hadley Centre Climate Programme (GA01101). HDG was supported by a Marie Curie Career Integration Grant from the European Commission. JP is supported by the German Research Foundation's Emmy Noether Program (PO 1751/1-1).

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
