# Peer review of "The C4MIP experimental protocol for CMIP6"

_Geoscientific Model Development, 2016_

## Short Comment (SC1) · 13 Apr 2016

Dear authors,

In agreement with the CMIP6 panel members, the Executive editors of GMD would like to establish a common naming convention for the titles of the CMIP6 experiment description papers.

The title of CMIP6 papers should include both the acronym of the MIP, and CMIP6, so that it is clear this is a CMIP6-Endorsed MIP.

Additionally, we strongly recommend to add a version number to the MIP description. The reason for the version numbers is so that the MIP protocol can be updated later, normally in a second short paper outlining the changes. See, for example: http://www.geosci-model-dev.net/special_issue11.html,

Good formats for the title include:
'XYZMIP (v1.0) contribution to CMIP6: Name of project'

or

'Name of Project (XYZMIP v1.0) contribution to CMIP6'

If you want to include a more descriptive title, the format could be along the lines of,

'XYZMIP (v1.0) contribution to CMIP6: Name of project - descriptive title'

or

'Name of Project (XYZMIP v1.0) contribution to CMIP6: descriptive title.'

When you revise your manuscript, please consider adding a version number to the title of your manuscript.

Yours,

Astrid Kerkweg

―――――――――――――――――――

---

## Referee Comment (RC1) · Anonymous Referee #1 · 14 Apr 2016

This is a critical manuscript laying out the criteria for a broad community Earth system model inter-comparison project addressing carbon cycling in both land and ocean systems to inform the next IPCC report. The authors provide historical context for the proposed experiment design, modification based on previous efforts, and detailed practical instruction for carrying out the inter-comparison.

I have minor suggestions noted below but otherwise find the manuscript to be a carefully considered continuation of previous efforts. Historically C4MIP has had a high impact on the scientific community and I expect this to continue based on this manuscript.

Details: P1L27 "...the design and documentation of individual simulations has been devolved to individual climate science communities." It's not clear what you mean by this ('devolved' is the word that's tripping me up), possibly reword?

Title & P1L29 While C4MIP is fairly widely known in the land carbon community it's

still not entirely self-explanatory and I, for one, frequently get it confused with the CMIP3/5/6 notation. I would like to see 'land carbon' somewhere in the title to make it a bit more explicit but would be open to other suggestions. The title is very acronym heavy.

P3L15 Nicely done recognizing that the number of experiments needs to be restricted due to computational challenges. I appreciate that the authors resisted the temptation to pile yet more runs into the design.

P3L29 Is there a citation for WCRP Grand Challenge?

P7L19, 21 TCR and TCRE are infrequently used in the manuscript. I suggest the authors consider writing the full names out to avoid cryptic acronyms as much as possible.

P7L28-31 I'm not entirely clear on the point of this paragraph. These two statements are relatively disjointed and need to be better integration into the section.

Figure2 I like this figure but if you need to cut figures I would cut this one. There seems to be a lot of careful treatment around concentration vs emissions forced which seems a tad unnecessary to me but I'm fine leaving it up to author's discretion on this. More pointedly, why are some of the lines solid and others dashed?

P21L28 There is a long space in this line.

P22L11 I don't believe that 'anomaly' is the word that you want in this line. Unless this is an American/British conflict, 'opposed' is more common here.

P25 I like how you address the soil carbon depth and fast/med/slow pool distinction here.

Figure 6: Please move the explanation of the colored arrows to the caption instead of stating in the main text.

P28 I appreciate the careful walk through discussing the connection between the tier 1 and 2 state and flux variables. Tedious as it is, it is necessary given my experience

with CMIP5. I look forward to the improvements this bodes for this next C4MIP round.

Section 4.2 Please link the variable names with their full description more explicitly. Though this is done with some variables (ex intPb) it is not done with all (ex FICR)

P40L10 Please give a bit more detail on the isotope reporting. The normalization factor could use more explanation.

―――――――――――――

---

## Referee Comment (RC2) · Anonymous Referee #2 · 15 Apr 2016

The past C4MIPs (e.g., that within CMIP5) have greatly stimulated research on carbon cycle-climate change feedbacks. The next C4MIP in association of CMIP6 is expected to continue such a great influence in the community.

Overall, this manuscript is well written and clearly describes C4MIP experiments and output requirements. If all participating modeling groups can follow the protocol, outputs of this MIP will be extremely useful for the community to learn more about the carbon cycle models in particular and consequently improve carbon cycle science in general.

I have no major criticisms on the manuscript but strongly suggest the protocol may consider require the modeling group to generate a pool-flux diagram for each model such as in Xia et al. 2013 with CABLE. Figs 5 and 6 already outline the pools and fluxes. The pool-flux diagram is a representation of Figs 5 and 6 for individual models and supposed to be corresponding to the matrix form of pools and fluxes. This protocol

may require modeling groups to report the pool-flux diagram and deposit their model codes in the Earth System Grid Federation (ESGF) or GitHUB. In this way, C4MIP will make the C4MIP modeling totally transparent for the community to learn about model structures, parameters, and output variables.

The description of outputs of modeled land carbon pools and fluxes is quite comprehensive but becomes quite lengthy. It will become much easier and clearer if the pools and fluxes are expressed in a matrix form as in Xia et al. 2013. Then all the elements in the matrix equation should be reported to allow accurate analysis of model outputs.

The sentence on page 26 "However, this distinction was not found useful by the community and as a result was not used in many analyses" may not be accurate. Distinguishing different soil pools is essential as repeatedly shown by many empirical and modeling studies. When you lump soil C together from many pools, it is almost impossible to understand how each model simulates soil carbon dynamics. I strongly recommend your protocol to require the report of outputs of individual soil carbon pools.

The same requirement should be make clear to report outputs of soil pools in different depths.

Other minor comments:

P. 6, L4-5, in addition to those differences in model structures, you may also set a goal to understand sciences behind model development, evaluation and improvement.

P.6, third para, we may bear in mind that most of the nitrogen models may not well reflect N processes in the real-world ecosystems as shown in some model intercomparison and model-data intercomparison studies. It requires transparency of models if we want to advance our field.

P.8, L1, should the subtitle 2.2.2 be "evaluation of global carbon cycle" or "evaluation of global carbon cycle models"?

P.9 second para, you may separately discuss evaluation techniques vs. datasets

P.9 third para, it appears that the idea on isotope modeling is not very well developed.

P. 11, C4MIP: How do you distinguish this C4MIP from other C4MIPs?

P. 14 L. 11-12, The sentence "A model cannot be conformant to the C4MIP protocol unless it can be run in both these configurations" is not very clear and needs more explanation.

P.16, L10. Land carbon cycle spin-up may use the semi-analytic method developed by Xia et al. 2012, which provides much more accurate estimates of steady states after spin-up. At least this protocol should recommend it.

P26. L11-17, Should report how soil carbon module is structured and all pools should be reported.

P28 L19, this is a great point to ensure mass conservation of C.

P28, L21, If all modeling groups use matrices to represent pools and fluxes, we will have a uniform way of reporting.

P27, l23-24 "Some models may also simulate this flux directly from vegetation to soil carbon, for instance, in the case of root exudates." In a matrix form, all those become straightforward.

P29, L9, for "in order to close the nitrogen budget", can you use the same language (i.e., mass conservation) as in the carbon cycle?

Page 30, L6 "to close nitrogen cycle budget over land" means the mass conservation but uses different terms.

P31, section 4.1.3. where in the paper did you specify the required outputs of the forcing variables?

---

## Short Comment (SC2) · 10 May 2016

General Comments

I found the paper well written and clear. I recommend it be accepted with minor revisions.

Specific Comments

1. Lines 1-5 Introduction – What are error bars on these carbon estimates? The values given have the units (i.e 1 PgC) appearing significant.

2. Page 6 and top of 7 – It is good to have a list of "coming attractions" for CMIP6. It would also be good to mention important things likely to be still missing – Very high ocean resolutions (10 km are finer), improvement in the way Land Use changes are being implemented in models, going away from the so-called big leaf vegetation models toward having multiple vegetation types in a grid cell, etc. Will the new features narrow

or increase the uncertainty of past and/or future estimate of carbon changes? What is the impact on missing processing on the uncertainty estimates for the future? I would like to read the authors' opinions on these questions.

3. Page 18, line 7 – "present" – Do you mean present or a given date (December 31, 2014 as an example). If the later, state the date and not use "present".

———————————————————————

---

## Referee Comment (RC3) · C. Huntingford (Referee) · 26 May 2016

The paper is a well thought-out plan for understanding and modelling the land and ocean roles in the carbon cycle that is being perturbed by human burning of fossil-fuels. It is an important paper, not just setting out a scientific research agenda, but also by informing future modelling protocols for climate-carbon cycle simulations. GMD is the appropriate journal for this manuscript. The diagrams help focus attention on what are the main issues still being developed in parameterised in the global carbon cycle.

The paper should be published and in its current form. Below are just a few small points that the authors might like to consider.

Happy to sign the review. Chris Huntingford

General

It might be worth mentioning directly that this paper partially addresses a frustration by

those who had built GCMs with a full interactive carbon cycle, and hence the logical approach was to force with different emissions scenarios. However to keep in GCM modelling groups who only describe the physical climate, then instead the CMIP5 protocol was to use prescribed forward profiles in atmospheric CO2 concentrations. There has been relatively little attempts to then find compatible emissions with those concentration trajectories. This CMIP6 protocol paper goes a long way to addressing that concern. Although concentration pathways will still be prescribed, the methodology in this paper submission explicitly states the need to back out land-atmosphere and ocean-atmosphere CO2 fluxes. From this, "permissible" emissions can be calculated. . . .Ah, reading on. . .. OK, can see p10, line 13 - and bullet point 5, page 11, that there will be emissions-driven simulations. Is this worth stating explicitly, point (3), line 8, page 2, up in the Abstract/Introduction, that the rcp-approach frustration is in part removed?.

P5. I always think the sentence (that appears, similarly, elsewhere too): "All models agreed qualitatively that the sign of the carbon-climate feedback was positive" should be given more context to those not so familiar with this area of work. This is not saying that the land and oceans automatically put more CO2 back in to the atmosphere under climate change, and the Introduction makes this clear. Would it be an idea to say something like: the direct effect of climate offsets some of the fertilisation-induced ability to draw-down atmospheric CO2?

Section 2.2 sets out three main expected scientific advances. This makes the paper very interesting beyond just protocol description. Maybe highlight these three paragraphs better, with either subheadings or short introductory sentences. E.g. "Terrestrial Nitrogen Modelling", "Enhanced Soil Modelling" or "Better Ocean Circulation Modelling".

Sorry if I've missed this somewhere tucked in the paper, but how is the emissions timeseries determined for the SSP5-8.5 scenario? The name hints this will be a scenario that gets an atmospheric concentration profile similar to the rcp85 prescription of CO2

concentration. Are "harmonised" emissions taken from Malte Meinhausen's rcp page? Here: http://www.pik-potsdam.de/∼mmalte/rcps/ (OK, there is a brief mention line 29, page 19, but no reference given).

P21. N-deposition. I had to run through this a couple of times to understand how the future scenarios of N deposition for SSP5-8.5 will be determined. Line 13, p 22 says: "The provided N-deposition data will cover both land and ocean". This looks like the preferred option – is there a reference to the model that will produce these fields, and the scenario used for the future? I wouldn't want to slow this paper down, so only if the authors have the time. But it would be helpful – especially for people coming to parts of the Earth System they are unfamiliar with – if all variables could have their units specified (e.g. myself, more familiar with the land surface, so units would help me better understand Figure 13). It could also avoid confusion when the CMIP5 .nc files are built – e.g. are flux units best saved as: /sec, /day or /hour.

Small things

Some extra keywords might help. "Global Carbon Cycle", "Climate Change", "Nitrogen Cycle"...

P6, top line – give the webaddress as a reference to the COP21 meeting.

https://unfccc.int/resource/docs/2015/cop21/eng/l09r01.pdf

Page 7, line 14. What is the difference between B and beta - and similarly between the two gammas.

Page 7. Is there a reference to the 1pctCO2 experiment. Or state that this is a cumulative increase – i.e. 1% per annum, year on year. Correct?\

Page 7, line 23. Typo: "provides"

P8, line 25. Is there any merit in mentioning that there are now a few Earth Observing datasets that can help constraint terrestrial land-atmospheric CO2 fluxes? (MODIS I

think, for NPP?).

Page 9, line 12. If we knew the fluxes well, and the residence times, then wouldn't that also give us the stores?

Page 9, paragraph on carbon-14 from nuclear weapons testing. I guess we know the magnitude and timing of the drivers of these well enough to make the simulations, given the secrecy of the cold war? Otherwise, we could suggest a detection-attribution style study is needed. So a bit like inclusion of volcanic eruptions in and D&A analysis.

P10. Is there a land surface MIP, that might mirror OMIP?

P21, line 23 – typo – extra white space

P23, line 15, type – "described".

Figure 5. What is the top-left box "(co2)"?

Figure 5. The line from the main box to "cProduct" (for products, e.g. "furniture") comes from cLitter. Shouldn't that box be linked to cVeg? The IKEA bookcases in my flat, they look like they were built from trees ("cVeg"), rather than their litterfall!

Page 26, line12. Is there a reason why the fast, medium and slow definitions were not used by the community, if this is actually the way carbon passes through separate pools?

Could be problems with the toner in my printer, but for Figure 6, the brown, yellow and orange colours are difficult to differentiate between. Maybe use red, yellow, green (and blue) colours? (Similarly Fig 8)

Maybe it's obvious, but "c" in variable names is for carbon pools and "f" for fluxes (e.g. Eqns on page 29). Might be worth just stating that.

More importantly on notation, are all the names in the Equations – e.g. on page 25, 26, 28, 29 – these are the specific names that will be used on the CMIP6 database.

This might be worth stressing if it is correct (so like "tas" is near-surface temperature in the CMIP5 database). That will aid 1-1 correspondence between the database and this manuscript.

As the quantities from this paper will be cited heavily, please use equation numbers at each point. I realise paper etiquette is to only number equations if they are explicitly cited in the main text, but here an exception could be made given the documentation implications of this paper.

The paper style change slightly around p33, where the physical state variables are presented more in bullet-point format. OK?

Conclusions. Although this is about the CMIP6 model setup and protocols, the paper is still also important in general terms, as it expresses current thinking in modelling climate and associated global geo-chemical cycles. Would it be appropriate to have a couple of sentences that outline what is still missing? So hinting at CMIP7. One key example might be the lack of a fully interactive methane cycle.

---

## Author Comment (AC1) · 9 Jun 2016

**Review comments in BLACK**

*Author responses in Blue/Italics*

Dear authors,

In agreement with the CMIP6 panel members, the Executive editors of GMD would like to establish a common naming convention for the titles of the CMIP6 experiment description papers.

The title of CMIP6 papers should include both the acronym of the MIP, and CMIP6, so that it is clear this is a CMIP6-Endorsed MIP.

Additionally, we strongly recommend to add a version number to the MIP description. The reason for the version numbers is so that the MIP protocol can be updated later, normally in a second short paper outlining the changes. See, for example:

http://www.geosci-model-dev.net/special_issue11.html,

Good formats for the title include:

'XYZMIP (v1.0) contribution to CMIP6: Name of project'

or

'Name of Project (XYZMIP v1.0) contribution to CMIP6'

If you want to include a more descriptive title, the format could be along the lines of,

'XYZMIP (v1.0) contribution to CMIP6: Name of project - descriptive title'

or

'Name of Project (XYZMIP v1.0) contribution to CMIP6: descriptive title.'

When you revise your manuscript, please consider adding a version number to the title of your manuscript.

Yours,

Astrid Kerkweg

*Thanks for this useful suggestion. Our title was already close to this, so we suggest a small revision as follows. Reviewer 1 also asked to clarify the acronyms in the title. We felt though that we did not need a version number as aligning this with CMIP6 is effectively a version number. To add one*

*on top of this would be confusing. We will add a sentence to the introduction to explain that this is our 3$^{rd}$ generation of C4MIP (following Friedlingstein et al 2006, and CMIP5).*

*new title:*

*C4MIP – The Coupled Climate-Carbon Cycle Model Intercomparison Project: Experimental protocol for CMIP6.*

*Text added to introduction:*

*"This is the third generation of C4MIP following the first coordinated experiments described in Friedlingstein et al. (2006) and the carbon cycle simulations which formed part of CMIP5 (Taylor et al., 2012)."*

---

## Author Comment (AC2) · 9 Jun 2016

**Review comments in BLACK**

*Author responses in Blue/Italics*

This is a critical manuscript laying out the criteria for a broad community Earth system model inter-comparison project addressing carbon cycling in both land and ocean systems to inform the next IPCC report. The authors provide historical context for the proposed experiment design, modification based on previous efforts, and detailed practical instruction for carrying out the inter-comparison.

I have minor suggestions noted below but otherwise find the manuscript to be a carefully considered continuation of previous efforts. Historically C4MIP has had a high impact on the scientific community and I expect this to continue based on this manuscript.

*Thank you for these supportive words*

Details: P1L27 ". . .the design and documentation of individual simulations has been devolved to individual climate science communities." It's not clear what you mean by this ('devolved' is the word that's tripping me up), possibly reword?

*We have reworded this to "delegated"*

Title & P1L29 While C4MIP is fairly widely known in the land carbon community it's still not entirely self-explanatory and I, for one, frequently get it confused with the CMIP3/5/6 notation. I would like to see 'land carbon' somewhere in the title to make it a bit more explicit but would be open to other suggestions. The title is very acronym heavy.

*As per this and also advice from the editor we have changed the title to better explain the acronym C4MIP. We note though that C4MIP is explicitly global (land and ocean) and coupled and not land-only. There is a land MIP (LS3MIP) and an activity of the Global Carbon Project called TRENDY that cover land-only simulations. C4MIP deals with the fully coupled climate-carbon cycle system.*

*new title:*

*C4MIP – The Coupled Climate-Carbon Cycle Model Intercomparison Project: Experimental protocol for CMIP6.*

P3L15 Nicely done recognizing that the number of experiments needs to be restricted due to computational challenges. I appreciate that the authors resisted the temptation to pile yet more runs into the design.

*Thank you. Although at a recent planning meeting we decided to add an additional (tier-2) simulation to look at carbon cycle feedbacks in an overshoot scenario. So in parallel to ScenarioMIP's SSP5-3.4-OS we now request a biogeochemically coupled version of that scenario. This will only add an extra 60 years (plus optional 200 if extended to 2300). We still feel that our experiment set is very compact and each simulation has a distinct and important application.*

P3L29 Is there a citation for WCRP Grand Challenge?

*The proposed grand challenge has now been endorsed by WCRP and the text updated to document this. It has been renamed as "carbon cycle" to narrow the scope from the initial proposal of "biogeochemical cycles". We cite the WCRP web page, but welcome the editor's advice on whether this is necessary or not:*

**http://www.wcrp-climate.org/grand-challenges**

P7L19, 21 TCR and TCRE are infrequently used in the manuscript. I suggest the authors consider writing the full names out to avoid cryptic acronyms as much as possible.

*We agree these are not used extensively in the manuscript and will spell out on each use*

P7L28-31 I'm not entirely clear on the point of this paragraph. These two statements are relatively disjointed and need to be better integration into the section.

*We have added a sentence to explain this and join up the two ideas presented here:*

*"C4MIP will use partially coupled simulations to isolate and quantify the sensitivity of carbon cycle components to climate and $CO_2$ separately and also the potentially large non-linear combination of these two components (Gregory et al., 2009; Schwinger et al., 2014). Simulations with only carbon cycle model components experiencing rising $CO_2$ (BGC-coupled) and the radiation components seeing the $CO_2$ rise (RAD-coupled) are used to quantify the carbon-concentration and carbon-climate feedbacks. Spatial patterns of these metrics can also be calculated (e.g. Roy et al., 2011, or Fig. 6.22 of the last IPCC WG1 assessment report Ciais et al. 2013) to establish areas of model agreement or disagreement."*

Figure2 I like this figure but if you need to cut figures I would cut this one. There seems to be a lot of careful treatment around concentration vs emissions forced which seems a tad unnecessary to me but I'm fine leaving it up to author's discretion on this. More pointedly, why are some of the lines solid and others dashed?

*We have not been asked to reduce overall length so decided to keep this figure. We have added to the caption that:*

*"Solid arrows depict internal data flow within the model, dashed arrows depict data output from the model."*

P21L28 There is a long space in this line.

*Thank you - removed*

P22L11 I don't believe that 'anomaly' is the word that you want in this line. Unless this is an American/British conflict, 'opposed' is more common here.

*We did mean anomaly in the context that these are added on top of the existing pre-industrial fields (so not instead of). Text has been clarified:*

*"… it is preferable to use the provided fields as anomalies which should be added to the ESM's pre-industrial N deposition fields."*

P25 I like how you address the soil carbon depth and fast/med/slow pool distinction here.

*Thank you. On suggestions from other reviews and discussion with colleagues we have added an additional (tier-2) data request for model to output more detail of their soil carbon pools. This will aid tracking and diagnosing turnover times for each model without having to assume a common structure.*

Figure 6: Please move the explanation of the colored arrows to the caption instead of stating in the main text.

*We agree and have done so*

P28 I appreciate the careful walk through discussing the connection between the tier 1 and 2 state and flux variables. Tedious as it is, it is necessary given my experience with CMIP5. I look forward to the improvements this bodes for this next C4MIP round.

*Thank you. We agree this is an important aspect to be very clear about*

Section 4.2 Please link the variable names with their full description more explicitly. Though this is done with some variables (ex intPb) it is not done with all (ex FICR)

*Thank you – we have checked through.*

P40L10 Please give a bit more detail on the isotope reporting. The normalization factor could use more explanation.

*We have added more detail on isotope reporting:*

*"Stocks and fluxes of carbon-14 should be normalized with the standard $^{14}C/C$ ratio, Rs, of 1.176 x $10^{-12}$ (Karlen et al. 1968). This means that reported stocks and fluxes of carbon-14 should be divided by Rs."*

---

## Author Comment (AC3) · 9 Jun 2016

**Review comments in BLACK**

*Author responses in Blue/Italics*

The past C4MIPs (e.g., that within CMIP5) have greatly stimulated research on carbon cycle-climate change feedbacks. The next C4MIP in association of CMIP6 is expected to continue such a great influence in the community.

Overall, this manuscript is well written and clearly describes C4MIP experiments and output requirements. If all participating modelling groups can follow the protocol, out- puts of this MIP will be extremely useful for the community to learn more about the carbon cycle models in particular and consequently improve carbon cycle science in general.

*Thank you for these supportive words*

I have no major criticisms on the manuscript but strongly suggest the protocol may consider require the modelling group to generate a pool-flux diagram for each model such as in Xia et al. 2013 with CABLE. Figs 5 and 6 already outline the pools and fluxes. The pool-flux diagram is a representation of Figs 5 and 6 for individual models and supposed to be corresponding to the matrix form of pools and fluxes. This protocol may require modeling groups to report the pool-flux diagram and deposit their model codes in the Earth System Grid Federation (ESGF) or GitHUB. In this way, C4MIP will make the C4MIP modeling totally transparent for the community to learn about modelstructures, parameters, and output variables.

*We agree that such a documentation of each model is extremely useful. We don't feel able to require all groups to follow the Xia reporting protocol exactly, but have suggested this as a way that could be used. We have added a request in the land-diagnostic section that all models document and report their structure and also report more detail of soil carbon pools above and below 1m.*

*The protocol that is currently specified as tier-1 in the manuscript allows analysis to compute the results as a 2x2 matrix (where the two members are live vegetation and dead CWD/litter/soil) as in Koven et al Biogeosciences 2015. Anything beyond that level of complexity requires assumptions that may not be shared universally (such as different structures/numbers of veg/CWD/litter/soil pools? is each PFT a separate matrix element? how to handle size-structured wood pools? how to handle vertically-resolved soil C? how to handle nonlinear soil C models? etc). We agree that this is a useful way of framing the issue, but do not feel the approach is yet ready to be mandated universally for all models.*

*Added text: "For CMIP5 the soil carbon pool was requested to be divided into components with fast, medium and slow turnover timescales. However, this distinction was not found useful by the community and as a result was not used in many analyses. For CMIP6, we are requesting a*

**breakdown in two different ways. Firstly based on the vertical distribution of soil carbon: total soil carbon should be split into above and below 1m depth (cSoil and cSoilBelow1m, respectively). Models which do not explicitly represent a vertical distribution of soil carbon should report all of their soil carbon in cSoil. The rationale for requesting this is the availability of several observation-based datasets that report soil organic matter content to 1 m depth. A second level of vertical reporting is then requested for tier 2. We ask that models with a vertical structure to their soil carbon report total soil carbon for each soil layer. As the structure for this may vary between models it is essential that the model is thoroughly documented.**

**In order to be able to diagnose and evaluate the turnover rates of carbon within the terrestrial system we make a second tier 2 request to report individual soil carbon pools. In order for this to be useful it is also required to report the turnover rate (defined as 1/residence time) for each pool. For models with a vertical structure we recognise that this could become an unmanageable number of pools reported, so we request that the individual pools are aggregated above and below 1m. To ensure such output is used correctly in analyses it is also essential that the pool-flux structure of each model is fully documented in its model description paper. This output will enable reduced complexity approaches (e.g. Xia et al., 2013) to recreate and analyse the soil carbon dynamics within each model."**

The description of outputs of modeled land carbon pools and fluxes is quite comprehensive but becomes quite lengthy. It will become much easier and clearer if the pools and fluxes are expressed in a matrix form as in Xia et al. 2013. Then all the elements in the matrix equation should be reported to allow accurate analysis of model outputs.

*As above (and as commented by review 1) we feel the length and comprehensiveness of documentation here is important and appropriate.*

The sentence on page 26 "However, this distinction was not found useful by the community and as a result was not used in many analyses" may not be accurate. Distinguishing different soil pools is essential as repeatedly shown by many empirical and modeling studies. When you lump soil C together from many pools, it is almost impossible to understand how each model simulates soil carbon dynamics. I strongly recommend your protocol to require the report of outputs of individual soil carbon pools.

*We believe the sentence to be true that very little use was made of these distinctions from the previous set of outputs (CMIP5). We do agree though that individual soil carbon pools should be reported and have updated the manuscript and our data request to reflect this (as per response to your point above).*

The same requirement should be make clear to report outputs of soil pools in different depths.

*We agree – we have now requested models to output all their pools individually above and below 1m.*

Other minor comments:

P. 6, L4-5, in addition to those differences in model structures, you may also set a goal to understand sciences behind model development, evaluation and improvement.

*We agree – it is very much our intention to stimulate such advances – especially with evaluation. This is explicit in our stated goals for C4MIP in the introduction*

P.6, third para, we may bear in mind that most of the nitrogen models may not well reflect N processes in the real-world ecosystems as shown in some model intercomparison and model-data intercomparison studies. It requires transparency of models if we want to advance our field.

*We agree that nitrogen, as well as many other ecosystem processes, are not always well represented. Again we have set evaluation of the models as a research priority.*

P.8, L1, should the subtitle 2.2.2 be "evaluation of global carbon cycle" or "evaluation of global carbon cycle models"?

*Thanks. We have corrected this to say "models"*

P.9 second para, you may separately discuss evaluation techniques vs. datasets

*We agree, but feel they are sufficiently linked that we have kept this paragraph as it is.*

P.9 third para, it appears that the idea on isotope modeling is not very well developed.

*It is true that coupled (GCM) modelling of isotopes is certainly not well developed. We hope that having an explicit request for groups to simulate carbon isotopes where possible and report the outputs will stimulate research in this area. Our analysis plans will evolve depending on how many ESMs are able to do so for CMIP6. This paragraph was revised for clarification.*

P. 11, C4MIP: How do you distinguish this C4MIP from other C4MIPs?

*This is the 3rd generation of C4MIP having evolved from the single publication of Friedlingstein et al (2006) that documented the first C4MIP intercomparison to the J. Climate special issue from CMIP5 carbon cycle analyses. We decided against applying a "vn3" label to this, but link it instead simply to CMIP6. This is now discussed in the introduction*

P. 14 L. 11-12, The sentence "A model cannot be conformant to the C4MIP protocol unless it can be run in both these configurations" is not very clear and needs more explanation.

*We have rephrased this for clarity – a model needs to be able to perform both emissions-driven and concentration-driven simulations in order to fulfil the C4MIP required set of simulations:*

*"A model must be able to run in both these configurations in order to perform the C4MIP simulations".*

P.16, L10. Land carbon cycle spin-up may use the semi-analytic method developed by Xia et al. 2012, which provides much more accurate estimates of steady states after spin-up. At least this protocol should recommend it.

*Thank you. We have added this to the list of possible techniques groups may choose from*

P26. L11-17, Should report how soil carbon module is structured and all pools should be reported.

*As above - We agree and request this now*

P28 L19, this is a great point to ensure mass conservation of C.

*thank you*

P28, L21, If all modeling groups use matrices to represent pools and fluxes, we will have a uniform way of reporting.

*As above we mention this approach as an option*

P27, l23-24 "Some models may also simulate this flux directly from vegetation to soil carbon, for instance, in the case of root exudates." In a matrix form, all those become straightforward.

*As above we mention this approach as an option*

P29, L9, for "in order to close the nitrogen budget", can you use the same language (i.e., mass conservation) as in the carbon cycle?

*We explicitly added mention of conservation*

Page 30, L6 "to close nitrogen cycle budget over land" means the mass conservation but uses different terms.

*We explicitly added mention of conservation*

P31, section 4.1.3. where in the paper did you specify the required outputs of the forcing variables?

*We do not request outputs of the forcing data*

---

## Author Comment (AC4) · 9 Jun 2016

**Review comments in BLACK**

*Author responses in Blue/Italics*

General Comments

I found the paper well written and clear. I recommend it be accepted with minor revisions.

*Thank you*

Specific Comments

1. Lines 1-5 Introduction – What are error bars on these carbon estimates? The values given have the units (i.e 1 PgC) appearing significant.

*These are reported to the nearest 5 PgC as per IPCC AR5, WG1 Ch.6 and Le Quere et al. we have now added in the uncertainty estimates also:*

*"Over the industrial era since about 1750, it is estimated that cumulative anthropogenic carbon emissions from fossil fuels and cement (405±20 PgC) and land use change (190±65 PgC) have been partitioned between the atmosphere (255±5 PgC), the ocean (170±20 PgC), and the terrestrial biosphere (165±70 PgC) (values to the nearest 5 PgC, from Le Quéré et al., 2015)."*

2. Page 6 and top of 7 – It is good to have a list of "coming attractions" for CMIP6. It would also be good to mention important things likely to be still missing – Very high ocean resolutions (10 km are finer), improvement in the way Land Use changes are being implemented in models, going away from the so-called big leaf vegetation models toward having multiple vegetation types in a grid cell, etc. Will the new features narrow or increase the uncertainty of past and/or future estimate of carbon changes? What is the impact on missing processing on the uncertainty estimates for the future? I would like to read the authors' opinions on these questions.

*All these ideas would be nice to pursue. We mention some of the key areas we think might change but feel it is premature to try to predict in more detail how some of the expected model changes will feed through to changes in results. For nitrogen there is evidence (from modelling and theory) of how this may affect outputs and so we feel that speculating on the sign (but not magnitude) of this response is appropriate. We have added that we would expect inclusion of permafrost carbon to also increase the carbon release due to climate warming – this is well founded and based on IPCC AR5 assessment:*

*Added to p.7: "AR5 assessed that permafrost carbon release was likely, and therefore would increase the climate-carbon cycle feedback, but with low confidence in the magnitude (Ciais et al., 2013)"*

*For other processes – such as enhanced ocean resolution we do not know how this may affect either baseline simulations nor the models' sensitivities to changes.*

*Improved treatment of land-use is also expected for CMIP6 but we leave discussion of this to LUMIP.*

*We also felt that the paper is already long and have tried not to increase the length through discussion – that may be more appropriate in another forum than the GMD documentation paper. We note that an increase in model SPREAD is not the same as increasing the uncertainty – it may simply be that models were artificially close to start with due to a common missing process. By representing this process, the spread of results may increase to better characterise the true and existing uncertainty.*

3. Page 18, line 7 – "present" – Do you mean present or a given date (December 31, 2014 as an example). If the later, state the date and not use "present".

*Thank you. This was sloppy and we meant the end of the CMIP6 historical period, defined as end of 2014. We have corrected this. Ditto in section 3.3.1.*

---

## Author Comment (AC5) · 9 Jun 2016

Author response to: RC3: review from Chris Huntingford

**Review comments in BLACK**

*Author responses in Blue/Italics*

**Review of paper: "The C4MIP experimental protocol for CMIP6" by Chris Jones et al.**

The paper is a well thought-out plan for understanding and modelling the land and ocean roles in the carbon cycle that is being perturbed by human burning of fossil-fuels. It is an important paper, not just setting out a scientific research agenda, but also by informing future modelling protocols for climate-carbon cycle simulations. GMD is the appropriate journal for this manuscript. The diagrams help focus attention on what are the main issues still being developed in parameterised in the global carbon cycle.

The paper should be published and in its current form. Below are just a few small points that the authors might like to consider.

Happy to sign the review. Chris Huntingford

*We thank Chris for his thoughtful comments and review*

**General**

It might be worth mentioning directly that this paper partially addresses a frustration by those who had built GCMs with a full interactive carbon cycle, and hence the logical approach was to force with different emissions scenarios. However to keep in GCM modelling groups who only describe the physical climate, then instead the CMIP5 protocol was to use prescribed forward profiles in atmospheric $CO_2$ concentrations. There has been relatively little attempts to then find compatible emissions with those concentration trajectories. This CMIP6 protocol paper goes a long way to addressing that concern. Although concentration pathways will still be prescribed, the methodology in this paper submission explicitly states the need to back out land-atmosphere and ocean-atmosphere $CO_2$ fluxes. From this, "permissible" emissions can be calculated. …Ah, reading on…. OK, can see p10, line 13 - and bullet point 5, page 11, that there will be emissions-driven simulations. Is this worth stating explicitly, point (3), line 8, page 2, up in the Abstract/Introduction, that the rcp-approach frustration is in part removed?.

*We have added mention of E-driven runs in the introduction:*

*"…by quantifying the role of carbon cycle feedbacks in the evolution of atmospheric $CO_2$ due to anthropogenic carbon emissions"*

P5. I always think the sentence (that appears, similarly, elsewhere too): "All models agreed qualitatively that the sign of the carbon-climate feedback was positive" should be given more context to those not so familiar with this area of work. This is not saying that the land and oceans automatically put more CO2 back in to the atmosphere under climate change, and the Introduction makes this clear. Would it be an idea to say something like: the direct effect of climate offsets some of the fertilisation-induced ability to draw-down atmospheric $CO_2$?

*Yes we agree this is important to get clear. This is actually already explained in the second half of that sentence:*

*"All models agreed qualitatively that the sign of the carbon-climate feedback was positive – i.e. the interaction of the carbon cycle with climate led to reduced carbon uptake..."*

Section 2.2 sets out three main expected scientific advances. This makes the paper very interesting beyond just protocol description. Maybe highlight these three paragraphs better, with either subheadings or short introductory sentences. E.g. "Terrestrial Nitrogen Modelling", "Enhanced Soil Modelling" or "Better Ocean Circulation Modelling".

*This is a nice idea, but when we discussed this we decided not to make this look like a prioritised research agenda, as these are only some example and cannot be exhaustive.*

Sorry if I've missed this somewhere tucked in the paper, but how is the emissions timeseries determined for the SSP5-8.5 scenario? The name hints this will be a scenario that gets an atmospheric concentration profile similar to the rcp85 prescription of $CO_2$ concentration. Are "harmonised" emissions taken from Malte Meinhausen's rcp page? Here:

http://www.pik-potsdam.de/~mmalte/rcps/

(OK, there is a brief mention line 29, page 19, but no reference given).

*Yes, this was mentioned briefly in the data section right at the end, but we will add a clearer explicit mention (in our section 3.3) of the forcing data, and references to the CMIP6 and forcing GMD papers and CMIP forcing web link:*

*Added to section 3.3: "The CMIP6 paper (Eyring et al., 2016) and a range of papers in the GMD CMIP6 special issue will document the forcings in more detail. The data will be made available from the CMIP6 webpage (http://www.wcrp-climate.org/wgcm-cmip/wgcm-cmip6)."*

P21. N-deposition. I had to run through this a couple of times to understand how the future scenarios of N deposition for SSP5-8.5 will be determined. Line 13, p 22 says: "The provided N-deposition data will cover both land and ocean". This looks like the preferred option – is there a reference to the model that will produce these fields, and the scenario used for the future?

*We have made this more explicit that the forcing for the CMIP6 scenarios will be provided elsewhere (and linked via the forcings web page as above). For C4MIP we only create our own idealised profile based on these spatial patterns.*

I wouldn't want to slow this paper down, so only if the authors have the time. But it would be helpful – especially for people coming to parts of the Earth System they are unfamiliar with – if all variables could have their units specified (e.g. myself, more familiar with the land surface, so units would help me better understand Figure 13). It could also avoid confusion when the CMIP5 .nc files are built – e.g. are flux units best saved as: /sec, /day or /hour.

*Thanks we will mention units where appropriate, but more importantly will cite the CMIP data request where all details (such as CF-NetCDF names, definitions and units) can be found:*

the WGCM Infrastructure Panel; https://www.earthsystemcog.org/projects/wip/

**Small things**

Some extra keywords might help. "Global Carbon Cycle", "Climate Change", "Nitrogen Cycle"…

*OK. We added "Global Carbon Cycle", "Climate Change". We felt that the paper was not explicitly about the global nitrogen cycle (any more than it is ocean ecosystems, permafrost, tropical forests etc etc) so we just added these two which apply generally to the whole of C4MIP.*

P6, top line – give the webaddress as a reference to the COP21 meeting.

https://unfccc.int/resource/docs/2015/cop21/eng/l09r01.pdf

*OK. Done.*

Page 7, line 14. What is the difference between B and beta  - and similarly between the two gammas.

*These refer to either FLUX or STORE definitions of the feedback terms (the former from Friedlingstein et al. and the latter from Arora et al.). In fact the symbols are not required for the text and we have removed them to avoid this confusion.*

Page 7. Is there a reference to the 1pctCO2 experiment. Or state that this is a cumulative increase – i.e. 1% per annum, year on year. Correct?

*Correct, yes. We now cite Eyring et al CMIP6 paper*

Page 7, line 23. Typo: "provides"

*Thanks - corrected*

P8, line 25. Is there any merit in mentioning that there are now a few Earth Observing datasets that can help constraint terrestrial land-atmospheric $CO_2$ fluxes? (MODIS I think, for NPP?).

*We provide some example of datasets (both EO and ground-based), but can't list them all. This specific  product is itself a numerical model to transform satellite radiances into an estimate of NPP, so we choose not to mention it explicitly.*

Page 9, line 12. If we knew the fluxes well, and the residence times, then wouldn't that also give us the stores?

*Yes, that's absolutely true, and is what we were trying to express – models have been evaluated well for fluxes but not residence times and hence the stocks are not constrained, so we need to step up and do both. The text has been clarified in this regard:*

*"In addition, consideration of residence times is crucial, which together with carbon fluxes jointly determine the stores."*

Page 9, paragraph on carbon-14 from nuclear weapons testing. I guess we know the magnitude and timing of the drivers of these well enough to make the simulations, given the secrecy of the cold war? Otherwise, we could suggest a detection-attribution style study is needed. So a bit like inclusion of volcanic eruptions in and D&A analysis.

*We have clarified that we are only requesting carbon isotopes be simulated in land and ocean model components. We are not requesting that models run carbon-14 or carbon-13 in emission-driven mode to simulate atmospheric D14C and d13C. We will provide atmospheric D14C and d13C forcing, which is well constrained by observations including tree rings, ice cores and direct measurements for 1850-2014. For future scenarios, we will provide atmospheric D14C and d13C forcing using a simple carbon cycle model following Graven, PNAS, 2015. Thus, the isotopes will primarily serve to provide tracers for the land and ocean components of the ESMs.*

*The reviewer is correct that one of the inputs needed for carbon cycle models to simulate atmospheric D14C is the production of carbon-14 from nuclear weapons testing. Bomb carbon-14 production has been estimated by Naegler, T., and I. Levin (2006), Closing the global radiocarbon budget 1945–2005, J. Geophys. Res.*

*We have revised the text on p9 to avoid confusion, and clarified the type of forcing for carbon-14 later in the document the specifications for isotopes.*

P10. Is there a land surface MIP, that might mirror OMIP?

*Good point – yes there is a land surface MIP called LS3MIP and an activity of the Global Carbon Project called TRENDY. Both focus on offline land-surface models, and the latter specifically on carbon. We cite them both here.*

P21, line 23 – typo – extra white space

*thanks. removed*

P23, line 15, type – "described".

*Thanks. Corrected*

Figure 5. What is the top-left box "(co2)"?

*This refers to atmospheric CO2 concentration. Whilst not a land-carbon pool, it is relevant here because we require it to be reported from the emissions-driven runs. The caption is updated to explain this.*

Figure 5. The line from the main box to "cProduct" (for products, e.g. "furniture") comes from cLitter. Shouldn't that box be linked to cVeg? The IKEA bookcases in my flat, they look like they were built from trees ("cVeg"), rather than their litterfall!

*Thanks. The figure is re-drawn to improve clarity. We were disappointed to hear that the reviewer has IKEA bookcases and not an antique Chippendale writing desk.*

Page 26, line12. Is there a reason why the fast, medium and slow definitions were not used by the community, if this is actually the way carbon passes through separate pools?

*We don't know why these were not well used, but now we request a finer breakdown of carbon pools from soil carbon models to avoid any ambiguities of re-classing them. See response to reviewer #2.*

Could be problems with the toner in my printer, but for Figure 6, the brown, yellow and orange colours are difficult to differentiate between. Maybe use red, yellow, green (and blue) colours? (Similarly Fig 8)

*Thanks. We agree it's important that these figures are clear both on screen and in print. We have darkened the brown arrows (now with white text) to improve the distinction. We have also passed these figures through a colour-blindness synthesiser to ensure they are accessible to anyone who may have difficulty with similar colours.*

Maybe it's obvious, but "c" in variable names is for carbon pools and "f" for fluxes (e.g. Eqns on page 29). Might be worth just stating that.

*Whilst not true of every variable (especially well known existing ones such as gpp) we have tried to do this for new ones. We have added mention of this:*

*"For ease of understanding we have adopted a convention for newly defined variables that a carbon pool is prefixed by a "c" (as in cVeg or cSoil) and a flux by an "f" (as in fLandToOcean). Some existing variables (e.g. gpp and npp) do not conform to this but are considered to be well known and do not need to be changed."*

More importantly on notation, are all the names in the Equations – e.g. on page 25, 26, 28, 29 – these are the specific names that will be used on the CMIP6 database. This might be worth stressing if it is correct (so like "tas" is near-surface temperature in the CMIP5 database). That will aid 1-1 correspondence between the database and this manuscript.

*Yes, this is the intention and we have checked through that this is the case.*

As the quantities from this paper will be cited heavily, please use equation numbers at each point. I realise paper etiquette is to only number equations if they are explicitly cited in the main text, but here an exception could be made given the documentation implications of this paper.

*Yes, this is a good idea and we have done so.*

The paper style change slightly around p33, where the physical state variables are presented more in bullet-point format. OK?

*These non-carbon variables are likely to be of more interest to other MIPs (notably LS3MIP) so we list them here for completeness but do not feel the same need for the detail given to the carbon cycle variables.*

Conclusions. Although this is about the CMIP6 model setup and protocols, the paper is still also important in general terms, as it expresses current thinking in modelling climate and associated global geo-chemical cycles. Would it be appropriate to have a couple of sentences that outline what is still missing? So hinting at CMIP7. One key example might be the lack of a fully interactive methane cycle.

*A fully interactive methane cycle would indeed be interesting, although this cuts across activities including C4MIP but also atmospheric chemistry (AerchemMIP). We have chosen to restrict discussion to CMIP6 advances – as CMIP7 is a very long way into the future. One MIP at a time!*

---

## Editor Comment (EC1) · C.A. Sierra (Editor) · 5 Jul 2016

I appreciate the comment made by the reviewer about a reference to the WCRP Grand Challenge. I think it's appropriate to include a reference, and the url included in the new version of the manuscript serves this purpose well.

---

## Author Response (AR1)

Friday 1st July, 2016

Dear Editor,

10 Please consider our revised submission to GMD following the review period. We have already addressed all of the review comments received in the open discussion and uploaded them to the discussion pages. Here we submit our revised manuscript.

In this document below is a track-changes version in accordance with our responses to the review comments. Attached as separate files are "clean" manuscript and abstract.

The biggest change is around adding clarity to our diagnostic outputs and requesting more detail of soil carbon pools (section 4.1.1). This was requested by multiple reviewers and we think it has significantly improved our experiment design. We have also added, with permission from the CMIP panel, a new

1

20 simulation: a biogeochemically coupled version of ScenarioMIP's overshoot experiment SSP5-3.4-Overshoot.

We hope the manuscript is now suited for publication in GMD.

25 Many thanks for considering our work,

Mrs Jones

Chris Jones On behalf of all co-authors

30

**The C4MIP - The Coupled Climate Carbon Cycle Model**

**Intercomparison Project:** experimental protocol for CMIP6**

Chris D. Jones1, Vivek Arora2, Pierre Friedlingstein3, Laurent Bopp4, Victor Brovkin5, John Dunne6, Heather Graven7, Forrest Hoffman8, Tatiana Ilyina5, Jasmin G. John6, Martin Jung9, Michio Kawamiya10, Charlie Koven11, Julia Pongratz5, Thomas Raddatz5, James T. Randerson12, Sönke Zaehle9

1Met Office Hadley Centre, Exeter, UK 2Canadian Centre for Climate Modelling and Analysis, Climate Research Division, Environment and Climate Change Canada

3College of Engineering, Mathematics and Physical Sciences, University of Exeter, Exeter, EX4 4QF, United Kingdom 10 4 Laboratoire des Sciences du Climat et de l'Environnement, LSCE/IPSL, CEA-CNRS-UVSQ, Université Paris-Saclay, F-91191 Gif-sur-Yvette, France 5Max Planck Institute for Meteorology

6NOAA/GFDL, Princeton, NJ, USA

7 Department of Physics and Grantham Institute, Imperial College London, UK 15 8Oak Ridge National Lab., TN, USA 9 Biogeochemical Integration Department, Max Planck Institute for Biogeochemistry, D-07745 Jena, Germany

10Japan Agency for Marine-Earth System-Science and Technology, Japan

11Earth Sciences Division, Lawrence Berkeley National Laboratory, Berkeley, California, USA

12Department of Earth System Science, University of California, Irvine, USA 20

Correspondence to: C.D. Jones (-chris.d.jones@metoffice.gov.uk)

Abstract. Coordinated experimental design and implementation has become a cornerstone of global climate modelling. Model Intercomparison Projects (MIPs) enable systematic and robust analysis of results across many models, by reducing

the influence of ad-hoc differences in model set-up or experimental boundary conditions. As it enters its 6th phase, the 25 Coupled Model Intercomparison Project (CMIP6) has grown significantly in scope with the design and documentation of individual simulations devolved delegated to individual climate science communities.

30

5

The Coupled Climate-Carbon Cycle Model Intercomparison Project (C4MIP) takes responsibility for design, documentation and analysis of carbon cycle feedbacks and interactions in climate simulations. These feedbacks are potentially large and play a leading order contribution in determining the atmospheric composition in response to human emissions of CO2 and in the setting of emissions targets to stabilise climate or avoid dangerous climate change. For over a decade C4MIP has coordinated coupled climate-carbon cycle simulations, and in this paper we describe the C4MIP simulations that will be

formally part of CMIP6. While the climate-carbon cycle community has created this experimental design, the simulations also fit within the wider CMIP activity, conform to some common standards including documentation and diagnostic requests and are designed to complement the CMIP core experiments known as the DECK.

- 5 C4MIP has 3 key strands of scientific motivation and the requested simulations are designed to satisfy their needs: (1) pre-industrial and historical simulations (formally part of the common set of CMIP6 experiments) to enable model evaluation;
   (2) idealised coupled and partially-coupled simulations with 1% per year increases in CO2 to enable diagnosis of feedback strength and its components; (3) future scenario simulations to project how the Earth System will respond to anthropogenic activity over the 21st century and beyond to anthropogenic activity.
- 10

This paper documents in detail these simulations, explains their rationale and planned analysis, and describes how to set-up and run the simulations. Particular attention is paid to boundary conditions, input data and requested output diagnostics. It is important that modelling groups participating in C4MIP adhere as closely as possible to this experimental design.

15 Keywords: Climate and Earth system modelling, CMIP6, Global Carbon Cycle, Climate Change

**1** Introduction**

Over the industrial era since about 1750, it is estimated that cumulative anthropogenic carbon emissions from fossil fuels and cement ( $405\pm20$  PgC) and land use change ( $190\pm65$  PgC) have been partitioned between the atmosphere ( $255\pm5$  PgC), the ocean ( $170\pm20$  PgC), and the terrestrial biosphere ( $165\pm70$ -
[revised manuscript text omitted]
                        | $\begin{array}{llllllllllllllllllllllllllllllllllll$                                                                          | C-driven                               | Fully coupled                     | 140                                                    | 1pctCO2 <del>cou</del> N de
p                                                                             |
| 1%BGC-Ndep                        | $\begin{array}{ccc} Idealised \ 1\% \ per \\ year \ CO_2 \ only, \\ BGC \ mode, \\ increasing \ N- \\ deposition \end{array}$ | C-driven                               | CO 2 affects BGC       | 140                                                    | lpctCO2 <del>bge</del> N de
p-bgc                                                                  |
| Hist/SSP5-8.5-
BGC             | Historical+SSP5-
8.5 up to 2300,
BGC mode                                                                               | C-driven                               | CO 2 affects BGC       | I. 155165
II. 85
III. 200                 | Historicalbge hist
-bgc , ssp5-85 -
bgc and ssp5-
85extbge85-
bgcExt |
| SSP5-3.4-
Overshoot-BGC | SSP5-3.4-OS up to
2300 in BGC
mode                                                                                      | C-driven                        | CO2 affects BGC | I. 60 (from
2040-
2100)
II. 200 | ssp534-over-bgc,
ssp534-over-
bgcExt                                                                          |
| CMIP DECK                         |                                                                                                                               |                                        |                                   |                                                        |                                                                                                                     |
| esm PIcontrol                     | pre-industrial
control run                                                                                                 | E-driven                               | Fully coupled                     | >200 as required
by CMIP DECK                       | EsmPIcontrol                                                                                                        |
| esm Historical                    | Historical                                                                                                                    | E-driven                               | Fully coupled                     | <del>155</del>                                         | esmHistorical                                                                                                       |

[revised manuscript text omitted]